# No More K-means:
# Single-Stage Sparse Coding for Efficient Multi-Vector Retrieval

**Lixuan Guo** [* 1 2]  **Yifei Wang** [* 3]  **Tiansheng Wen** [4 1]  **Aosong Feng** [5]  **Stefanie Jegelka** [6 7]  **Chenyu You** [1]

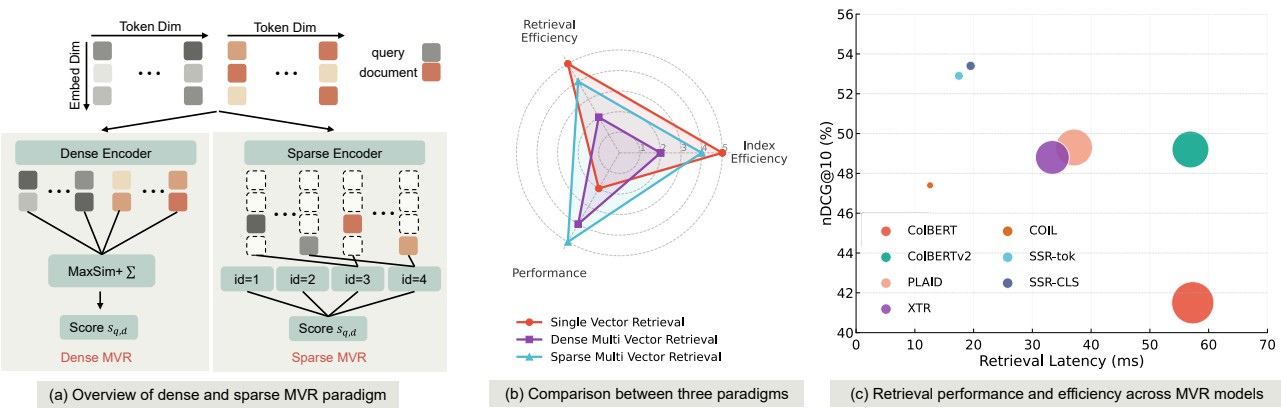

*Figure 1.* **Single-stage Sparse Retrieval (SSR).** (a) *Paradigm Comparison.* Unlike dense MVR, which compresses embeddings into low-dimensional vectors and requires exhaustive token-pair computations, our sparse MVR projects features into a high-dimensional sparse space via Sparse Autoencoders (SAE). This enables efficient interaction calculation solely on overlapping activated neurons. (b) *Trade-off Analysis.* Single Vector Retrieval offers high efficiency but suffers from semantic compression loss. Dense MVR preserves token-level details but incurs heavy computational overheads due to clustering and multi-stage pruning (inefficient indexing). SSR bridges this gap, accelerating retrieval via natural inverted indexing while retaining fine-grained semantic information. (c) *Performance vs. Efficiency.* SSR achieves state-of-the-art retrieval performance with halved retrieval latency compared to competitive baselines. Notably, it delivers a 15x reduction in indexing time (indicated by bubble size) by eliminating the clustering bottleneck.

## Abstract

Multi-vector retrieval (MVR) models, exemplified by ColBERT, have established new benchmarks in retrieval accuracy by preserving fine-grained token-level interactions. However, this granularity imposes prohibitive storage and retrieval efficiency bottlenecks: to manage the immense memory footprint and computational overhead of billion-scale token vectors, state-of-the-art systems are forced to rely on aggressive dimension reduction and complex clustering (e.g., K-means). This compromise introduces two critical limitations: excessive indexing latency of

clustering large-scale corpora and semantic information loss inherent to compression. In this paper, we propose *Single-stage Sparse Retrieval* (**SSR**), a paradigm shift that replaces expensive clustering with efficient sparse coding. Instead of compressing features into low-dimensional dense vectors, we utilize Sparse Autoencoder (SAE) to project token embeddings into a high-dimensional but highly sparse representation. This transformation enables us to bypass vector clustering entirely and leverage inverted indexing for precise, high-throughput retrieval. Extensive experiments on the BEIR benchmark demonstrate that SSR achieves a "trifecta" of improvements: it reduces indexing time by 15x compared to ColBERTv2, halves retrieval latency, and simultaneously improves retrieval performance over leading baselines.

---

[*]Equal contribution  [1]Stony Brook University [2]Xidian University [3]Amazon AGI SF Lab [4]Georgia Tech [5]Yale University [6]TUM [7]MIT. Correspondence to: Chenyu You <chenyu.you@stonybrook.edu>.

*Proceedings of the 43rd International Conference on Machine Learning*, Seoul, South Korea. PMLR 306, 2026. Copyright 2026 by the author(s).

# 1. Introduction

Neural information retrieval has long faced a fundamental trade-off between precision and efficiency (Khattab & Zaharia, 2020; Cao et al., 2023; Zhou et al., 2023; Santhanam et al., 2022a). While Single-Vector Retrieval (SVR) enables high-throughput search via simple dot products, it often struggles to capture the intricate semantic nuances of complex queries. In contrast, Multi-Vector Retrieval (MVR) paradigms, pioneered by ColBERT (Khattab & Zaharia, 2020), have established a new standard for effective retrieval. By preserving documents as sequences of token-level embeddings and employing late interaction (e.g., MaxSim) mechanisms, MVR achieves fine-grained semantic alignment that SVR typically misses.

However, such precision imposes substantial storage and computational overheads. Representing documents as bags of vectors causes the index size to explode by orders of magnitude compared to single-vector baselines. To make deployment feasible, state-of-the-art systems like PLAID (Santhanam et al., 2022a) rely on aggressive approximation strategies, such as Vector Quantization (VQ) and large-scale clustering (e.g., K-means). While these engineering optimizations mitigate retrieval latency, they leave two critical issues unresolved. First, compressing rich token embeddings into short codes or centroids inevitably incurs information loss, sacrificing the very semantic details MVR aims to preserve. Second, the indexing process remains prohibitively complex: performing clustering on billion-scale token datasets is computationally expensive, creating a severe bottleneck for index construction and real-time updates. This leads us to a pivotal question:

> *Can we retain the token-level granularity of MVR while achieving similar retrieval speed of lexical search, without the heavy burden of clustering?*

In this paper, we propose a paradigm shift from dense approximation to **S**ingle-stage **S**parse **R**etrieval (**SSR**). As shown in Figure 1 (**Left**), instead of compressing token embeddings into lower-dimensional dense embeddings, we project them into a significantly higher-dimensional but highly sparse feature space using Sparse Autoencoder (SAE) (Gao et al., 2024; Lee et al., 2006). This transformation disentangles complex token semantics into sparse vectors with few active neurons (e.g., 32). Crucially, this sparsity unlocks a structural advantage that dense methods lack: it enables us to discard K-means entirely and utilize the inverted index, allowing each active dimension to function as a "pseudo token", which is the same efficient data structure powering traditional keyword search (e.g., BM25 (Robertson et al., 1995)). We present two implementations: SSR-tok, which only focuses on token-level interactions, and SSR-CLS, which incorporates global semantics by integrating [CLS]

embedding similarities.

We empirically validate SSR on the MS MARCO (Nguyen et al., 2016) (in-domain) and BEIR (Thakur et al., 2021) (out-of-domain) benchmarks. Our results demonstrate that SSR simultaneously optimizes accuracy, speed, and scalability: it matches or exceeds the performance of state-of-the-art retrievers (achieving an average 2.2% improvement over the strongest baseline) while nearly halving retrieval latency and reducing index construction time by over 15× (by eliminating the clustering overhead). Furthermore, we demonstrate the scalability of our approach by applying it to modern Large Language Model backbones (Llama-Embed-8B (Babakhin et al., 2025)), achieving performance competitive with the latest single-vector embedding models. We discuss in detail the evolution of single-vector retrieval and multi-vector retrieval techniques in Appendix B, highlighting the correlations and limitations of existing methods that motivate SSR. Our code is released at `https://github.com/Y-Research-SBU/SSR`. Our contributions are as follows:

- We propose *Single-stage Sparse Retrieval*, a framework for efficient and effective multi-vector sparse coding, enabling direct use of inverted indices for single-stage semantic retrieval.

- We introduce a hybrid training objective that combines reconstruction loss with a multi-vector contrastive loss, ensuring that the learned sparse features are discriminative for ranking tasks.

- Extensive in-domain and out-of-domain evaluations confirm that SSR effectively improves the efficiency-effectiveness trade-off frontier, delivering sub-20ms retrieval latency and state-of-the-art indexing speed without compromising retrieval quality.

# 2. Background

## 2.1. Problem Formulation

The objective of a retrieval task is to get a ranked list of documents from a large-scale corpus $\mathcal{D} = \{D_1, \ldots, D_{|\mathcal{D}|}\}$ that are semantically relevant to a user query $Q$. Both the query $Q = (q_1, \ldots, q_{|Q|})$ and the document $D = (d_1, \ldots, d_{|D|})$ are typically represented as sequences of tokens. The core challenge is to learn a parameterized scoring function $s_\theta(Q, D) \in \mathbb{R}$ that quantifies the relevance between the query-document pair. Ideally, for a relevant document $D^+$ and a non-relevant document $D^-$, the model should satisfy $s_\theta(Q, D^+) > s_\theta(Q, D^-)$.

## 2.2. Existing Multi-Vector Retrieval Paradigm

Despite architectural variations in recent literature, the inference process of modern Multi-Vector Retrieval (MVR)

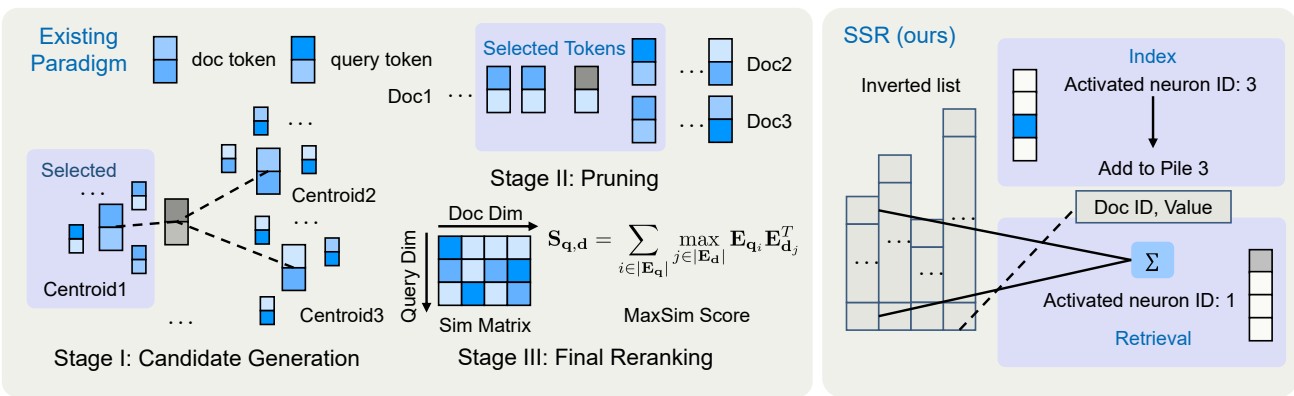

*Figure 2.* Conceptual comparison between standard retrieval paradigms (e.g., PLAID (Santhanam et al., 2022b)) and our proposed SSR.

models generally converges to a three-stage filter-and-refine paradigm, as illustrated in Figure 2. Formally, given a query $Q = \{q_1, \ldots, q_N\}$ and a document corpus $\mathcal{D}$, the goal is to efficiently identify top-$k$ documents that maximize the late interaction score $S(Q, D)$.

**Stage I: Indexing & Candidate Generation.** Handling billion-scale token vectors necessitates an efficient pre-filtering mechanism. Specifically, document tokens are mapped to a codebook of centroids $\mathcal{C} = \{\mathbf{c}_1, \ldots, \mathbf{c}_K\}$ via clustering engines such as FAISS (Douze et al., 2024) in the indexing stage. During retrieval, for each query token $q_i$, the system identifies a set of nearest centroids $\mathcal{C}_i \subset \mathcal{C}$ and retrieves the associated document list. The initial candidate set $\mathcal{D}_{\text{cand}}$ is the union of documents hit by these active centroids:

$$\mathcal{D}_{\text{cand}} = \bigcup_{q_i \in Q} \bigcup_{\mathbf{c} \in \mathcal{C}_i} \phi(\mathbf{c}) \tag{1}$$

where $\phi(\mathbf{c})$ represents the set of documents that include tokens in the centroid $\mathbf{c}$. This stage typically prunes the search space by filtering out $> 98\%$ of the original corpus.

**Stage II: Approximate Scoring & Pruning.** Performing fine-grained interaction on $\mathcal{D}_{\text{cand}}$ is still computationally prohibitive due to the high cost of memory access and floating-point operations. Therefore, this stage employs approximate scoring using compressed representations. For instance, PLAID (Santhanam et al., 2022a) and EMVB (Nardini et al., 2024) estimate the similarity using centroid-level interactions or bit-vectors rather than reconstructing the full embeddings. The approximate score $\hat{S}(Q, D)$ is often calculated as:

$$\hat{S}(Q, D) = \sum_{i=1}^{|Q|} \max_{j=1}^{|D|} (\mathbf{q}_i \cdot \mathbf{c}_{d_j}) \tag{2}$$

where $\mathbf{c}_{d_j}$ represents the centroid of the $j$-th document token $d_j$. Based on $\hat{S}$, the candidate set is aggressively pruned to

a much smaller subset $\mathcal{D}_{\text{top}} \subset \mathcal{D}_{\text{cand}}$ (typically thousands of documents).

**Stage III: Final Reranking with Decompression.** In the final stage, full-precision vectors are reconstructed to resolve minor semantic differences. Methods like ColBERTv2 (Santhanam et al., 2022b) utilize residual compression, where the reconstructed vector $\tilde{\mathbf{d}}_j \approx \mathbf{c}_{d_j} + \mathbf{r}_{d_j}$. The final ranking is determined by the precise MaxSim operation:

$$S(Q, D) = \sum_{i=1}^{N} \max_{j=1}^{M} (\mathbf{q_i} \cdot \tilde{\mathbf{d}}_j) \tag{3}$$

This ensures that the final top-ranked documents are selected based on the highest fidelity representation.

**Analysis: Efficiency Gain and Limitations.** The paradigm described above significantly reduces retrieval costs by shifting the heavy computation from the entire corpus to a progressively smaller candidate set. By utilizing quantization and approximate scoring, modern engines achieve sub-second latency for million-scale corpora. However, substantial challenges remain. **First**, the indexing process is computationally expensive and slow, primarily due to the overhead of clustering billion-scale tokens and computing residuals. **Second**, the multi-stage pruning pipeline introduces complex control logic and memory access patterns, where the overhead of decompression and repeated scoring can become a bottleneck. **Finally**, the MaxSim operation in the final stage, even with SIMD optimizations, remains theoretically quadratic with respect to sequence length (i.e., $O(N \times M)$), limiting the throughput for long-context retrieval.

## 3. New Paradigm: from Density to Sparsity

To address the efficiency limitations and indexing overhead inherent in the past multi-vector systems, we propose **Single-stage Sparse Retrieval (SSR)**. Instead of relying on ex-

pensive clustering and aggressive dimensionality reduction, SSR projects token embeddings into a high-dimensional, highly sparse feature space via Sparse Autoencoder (SAE), enabling direct and efficient semantic retrieval.

### 3.1. Sparse Late Interaction Scoring

The core of our SSR paradigm is to maintain the token-level granularity of late interaction while operating in a sparse latent space. Given a query $Q = \{q_1, ..., q_{|Q|}\}$ with $|Q|$ tokens and a document $D = \{d_1, ..., d_{|D|}\}$ with $|D|$ tokens, each token is first encoded into dense representation by backbone encoder such as BERT, and then mapped into a sparse vector $\mathbf{z} \in \mathbb{R}^h$ utilizing the learned SAE, after which the query is converted to $Q' = \{\mathbf{z}_{q_1}, \mathbf{z}_{q_2}, ..., \mathbf{z}_{q_{|Q|}}\} \in \mathbb{R}^{h \times |Q|}$ while document is converted to $D' = \{\mathbf{z}_{d_1}, \mathbf{z}_{d_2}, ..., \mathbf{z}_{d_{|D|}}\} \in \mathbb{R}^{h \times |D|}$. We adopt the MaxSim operator to calculate the fine-grained similarity. However, unlike dense retrieval, the interaction only occurs between activated neurons:

$$S(Q, D) = \sum_{i=1}^{N} \max_{j=1}^{M} \left( \sum_{u \in \mathcal{A}_K(\mathbf{z}_{q_i}) \cap \mathcal{A}_K(\mathbf{z}_{d_j})} \mathbf{z}_{q_i}^{(u)} \cdot \mathbf{z}_{d_j}^{(u)} \right) \tag{4}$$

where $\mathcal{A}_K(\cdot)$ denotes the set of indices for the top-$K$ largest neurons.

Appendix A provides a bounded-distortion analysis of the sparse scoring rule in Equation (4). Under small SAE reconstruction error and restricted decoder near-orthogonality on active sparse supports, the sparse inner product $\mathbf{z}_{q_i}^\top \mathbf{z}_{d_j}$ approximates the dense token similarity $\mathbf{q}_i^\top \mathbf{d}_j$ with bounded error. This token-level guarantee further extends through the MaxSim operator, showing that the full SSR late-interaction score remains close to the dense late-interaction score.

### 3.2. Hybrid Training of Sparse Projectors

To ensure that the sparse features are both reconstructive and discriminative for retrieval, we train two separate SAEs, with $\mathbf{E}_{\text{tok}}$ for regular tokens and $\mathbf{E}_{\text{[CLS]}}$ for global semantic tokens.

**Unsupervised TopK Sparse Autoencoding.** For feature $\mathbf{x} \in \mathbb{R}^d$, we apply the autoencoding process with encoder $\mathbf{W}_{\text{enc}} \in \mathbb{R}^{h \times d}$ to reach the targeted sparsity level $K$ with TopK operator and reconstruct original feature $\hat{\mathbf{x}}$ with decoder $\mathbf{W}_{\text{dec}} \in \mathbb{R}^{d \times h}$:

$$\mathbf{z} = \text{TopK}(\mathbf{W}_{\text{enc}}(\mathbf{x} - \mathbf{b}_{\text{pre}}) + \mathbf{b}_{\text{enc}}) \tag{5}$$

$$\hat{\mathbf{x}} = \mathbf{W}_{\text{dec}}\mathbf{z} + \mathbf{b}_{\text{pre}} \tag{6}$$

where $h$ is the hidden dimension, $\mathbf{b}_{\text{pre}}$ and $\mathbf{b}_{\text{enc}}$ are bias terms and the TopK operation sets all values to zero except the $K$ largest. The training goal is to minimize the difference

between original feature and the reconstructed feature under sparsity $\mathcal{L}_{\text{recon}}(k) = \|\mathbf{x} - \hat{\mathbf{x}}\|_2^2$ under sparsity $k$.

Following (Gao et al., 2024; Wen et al., 2025; Guo et al., 2026; Wang et al., 2024), we expand standard reconstruction loss with Multi-TopK loss, auxiliary loss $\mathcal{L}_{\text{aux}}$ and sparse contrastive loss and the overall loss is:

$$\mathcal{L}_{\text{unsup}} = \mathcal{L}_{\text{recon}}(k) + \frac{1}{8}\mathcal{L}_{\text{recon}}(4k) + \alpha\mathcal{L}_{\text{aux}}(k_{\text{aux}}) + \beta\mathcal{L}_{\text{cl}} \tag{7}$$

where $L_{\text{aux}}(k_{\text{aux}})$ calculates the reconstruction loss using the top-$k_{\text{aux}}$ neurons that have not been activated for a long time and sparse contrastive loss $L_{\text{cl}}$ encourages SAE to distinguish between positive and negative pairs with the following formula:

$$\mathcal{L}_{\text{cl}} = -\frac{1}{|\mathcal{B}|}\sum_{i=1}^{|\mathcal{B}|} \log \frac{e^{\mathbf{z}_i^T \mathbf{z}_i}}{e^{\mathbf{z}_i^T \mathbf{z}_i} + \sum_{j \neq i}^{|\mathcal{B}|} e^{\mathbf{z}_i^T \mathbf{z}_j}} \tag{8}$$

where $\mathcal{B}$ is the set of all tokens in the training sentence batch.

**Supervised Contrastive Learning.** Furthermore, we incorporate an additional supervised contrastive loss term to help Sparse Autoencoder capture the semantic difference between one query's positive and negative documents, which is followed by most mature dense MVR paradigms (Khattab & Zaharia, 2020; Gao et al., 2021; Lee et al., 2023):

$$\mathcal{L}_{\text{CE}} = -\log \frac{e^{\text{sim}(Q, D^+)}}{\sum_{D \in \mathcal{D}} e^{\text{sim}(Q, D)}} \tag{9}$$

where $Q$ is a query, $\mathcal{D}$ is a set of mini-batch documents with one positive document $D^+$ and $|\mathcal{D}| - 1$ negative documents $D^-$ for $Q$ and sim() is the similarity score that measures semantic similarity between two contexts. When trained on regular tokens, the similarity score is sum of MaxSim of each query token, while when trained on special token [CLS], the similarity score is cosine similarity.

**Overall Training Objective.** Finally, we optimize the sparse projector through a combination of unsupervised and supervised sparse learning, whose final training objective is formulated as:

$$\mathcal{L}_{SSR} = \mathcal{L}_{\text{unsup}} + \gamma\mathcal{L}_{\text{CE}} \tag{10}$$

where $\gamma$ is the hyperparameter to balance the two loss components and is set to 0.05 for default.

### 3.3. Sparsity-Enabled Efficient Indexing and Retrieval

Leveraging the high sparsity ($K \ll h$) of sparse features, we design a retrieval structure that circumvents the expensive cluster-based scanning of dense MVR. We first introduce the base retrieval paradigm (SSR) which utilizes neuron-level inverted indexing, followed by an accelerated variant (SSR++) employing a coarse-to-fine pruning strategy.

*Table 1.* **Comparative analysis of retrieval performance and efficiency against lightweight multi-vector and sparse lexical baselines**. All methods are evaluated under a controlled experimental setup. The best results are highlighted in **bold**, and the second-best are underlined. Retrieval performance is reported in nDCG@10, while efficiency is measured by per-query retrieval latency on the MSMARCO passage ranking dataset. Unless otherwise specified, the reported SSR-tok and SSR-CLS latency numbers use the SSR++ acceleration strategy described in Section 3.3.

| Method | Time | MS | AR | CF | CV | DB | FE | FQ | HQ | NF | NQ | QU | SD | SF | TO | Avg. |
|---|---|---|---|---|---|---|---|---|---|---|---|---|---|---|---|---|
| **Late-interaction dense retrieval** | | | | | | | | | | | | | | | | |
| ColBERT | 57.3 | 36.0 | 23.6 | 18.1 | 67.9 | 39.2 | 76.9 | 31.7 | 59.2 | 30.7 | 51.8 | 85.7 | 14.6 | 66.8 | 20.5 | 41.5 |
| COIL | 12.6 | 35.3 | 29.5 | 21.7 | 66.8 | 39.9 | **83.8** | 31.3 | **71.3** | 33.2 | 52.1 | 83.8 | 15.5 | 70.7 | 28.1 | 47.4 |
| ColBERTv2 | 56.9 | 39.8 | 46.3 | 17.3 | 73.5 | 44.9 | 78.7 | 36.1 | 66.6 | 33.5 | 56.3 | 85.3 | 15.2 | 69.1 | 26.1 | 49.2 |
| PLAID | 37.1 | 39.7 | 46.1 | 17.4 | 73.7 | 44.7 | 78.8 | 36.3 | 66.7 | 33.8 | 56.4 | 85.0 | 15.4 | 69.2 | 26.3 | 49.3 |
| AligneR | 51.8 | 38.8 | 28.9 | 18.2 | 68.8 | 41.6 | 72.4 | 33.5 | 61.5 | 34.0 | 52.4 | 82.3 | 14.3 | 70.3 | 34.8 | 46.6 |
| CITADEL | 15.7 | 39.9 | 51.1 | 18.3 | 71.5 | 42.2 | 76.6 | 33.0 | 66.3 | 33.7 | 54.0 | 85.3 | 15.9 | 69.0 | 34.2 | 49.4 |
| XTR | 33.4 | 45.0 | 40.7 | 20.8 | 73.6 | 40.8 | 73.7 | 34.8 | 64.7 | 34.0 | 52.8 | 85.9 | 14.6 | 71.1 | 31.3 | 48.8 |
| **Learned sparse retrieval** | | | | | | | | | | | | | | | | |
| Splade-v2 | 16.3 | 43.3 | 47.6 | 23.5 | 71.5 | 43.4 | 78.7 | 33.8 | 68.4 | 33.5 | 52.1 | 83.8 | 15.9 | 69.4 | **36.3** | 50.1 |
| Splade-v3 | 16.6 | 43.9 | **50.8** | 23.3 | 74.8 | 45.2 | 79.8 | 37.5 | 69.2 | 35.8 | 58.6 | 81.4 | 15.8 | 71.0 | 29.6 | 51.2 |
| **Late-interaction sparse retrieval** | | | | | | | | | | | | | | | | |
| SSR-tok | 17.5 | 45.2 | 46.1 | 22.8 | 76.4 | 48.2 | 81.9 | 39.4 | 69.5 | 38.8 | 60.7 | 88.3 | 17.5 | 72.9 | 33.1 | 52.9 |
| SSR-CLS | 19.5 | **45.5** | 46.6 | **23.5** | **76.8** | **48.4** | 82.8 | **40.0** | 69.9 | **39.1** | 61.2 | **88.7** | 17.9 | **73.3** | 33.7 | **53.4** |

**Neuron-Level Inverted Indexing.** Unlike dense methods that require clustering, we directly build inverted indices on active neurons. Let $\mathbf{z}_t \in \mathbb{R}^h$ denote the sparse representation of a document token $d$. For each neuron dimension $u \in \{1, ..., h\}$, we construct a posting list $\mathcal{I}_u$. To support the MaxSim operator efficiently, we store the maximum impact of neuron $u$ within a document $D$:

$$\mu_{D,u} = \max_{t \in D} \mathbf{z}_t^{(u)} \quad (11)$$

The posting list is thus defined as $\mathcal{I}_u = \{(D, \mu_{D,u}) \mid \mu_{D,u} > 0\}$. Furthermore, to facilitate retrieval, each list $\mathcal{I}_u$ is partitioned into fixed-size blocks, where each block $B$ stores an upper-bound score $U_B = \max_{D \in B} \mu_{d,u}$. This block-level organization enables early pruning during posting-list traversal, allowing SSR to avoid scoring documents whose maximum possible contribution is already insufficient for the current top-$K$ set.

**Single-stage Sparse Retrieval (SSR).** Given a query $Q = \{q_1, ..., q_N\}$, the goal of SSR is to compute the exact late-interaction score using the full set of activated neurons. For each query token embedding $\mathbf{q}_i$, we identify its top-$K$ activated neurons $\mathcal{A}_K(\mathbf{q}_i)$. The system traverses the union of posting lists corresponding to these neurons. Since the number of activated neurons $K$ is small (e.g., $K = 32$), this process is significantly faster than scanning dense vector clusters. However, traversing $N \times K$ lists can still be optimized for ultra-low latency scenarios.

**Accelerated Retrieval with Pruning (SSR++).** To further reduce latency, we propose SSR++, a coarse-to-fine pipeline that progressively prunes the search space.

**Step 1: Coarse Scoring.** For each query token $q_i$, we extract only the principal active neurons $\mathcal{A}_{K_{\text{coarse}}}(\mathbf{q}_i)$, where $K_{\text{coarse}} < K$ (we select $K_{\text{coarse}}$ as 4). We compute an approximate upper-bound score $\hat{S}_{\text{coarse}}$ by traversing only these principal posting lists:

$$\hat{S}_{\text{coarse}}(Q, D) = \sum_{i=1}^{N} \sum_{u \in \mathcal{A}_{K_{\text{coarse}}}(\mathbf{q}_i)} (\mathbf{q}_i^{(u)} \cdot \mu_{D,u}) \quad (12)$$

During this traversal, we utilize block upper-bounds $U_B$ to skip blocks that cannot contribute enough to exceed the current top-k threshold, rapidly producing a small candidate set $\mathcal{C}_1$.

**Step 2: Exact Refinement.** For the refined subset $\mathcal{C}_1$, we revert to the full set of activated neurons $\mathcal{A}_K(\mathbf{q}_i)$ to compute the precise late-interaction score as defined in Equation 4. This ensures that the final ranking captures the precise semantic alignment using all $K$ features while performing expensive computations on only a fraction of documents.

**Complexity Analysis.** The standard SSR scales with $O(N \cdot K \cdot \bar{L})$, where $\bar{L}$ is the average posting length. In contrast, SSR++ reduces the Stage I complexity to $O(N \cdot K_{\text{coarse}} \cdot L_{\text{skip}})$, where $K_{\text{coarse}}$ is significantly smaller than $K$ and $L_{\text{skip}} \ll \bar{L}$ due to block skipping. This efficiency gain is further supported by the bounded-distortion analysis in Appendix A, which shows that sparse active-neuron

*Table 2.* **Retrieval performance comparison when utilizing llama-embed-nemotron-8b (Babakhin et al., 2025) as backbone.** The maximum values are indicated in **bold**, while the second-highest values are underlined.

| Model | MS | AR | CF | CV | DB | FE | FQ | HQ | NF | NQ | QU | SD | SF | TO | Avg. |
|---|---|---|---|---|---|---|---|---|---|---|---|---|---|---|---|
| llama-8B | 61.7 | 75.7 | 44.9 | 89.2 | 40.1 | 93.5 | 61.4 | 76.7 | 45.1 | 66.2 | 88.1 | 28.2 | 82.0 | 68.8 | 65.8 |
| Qwen3-8B | **62.4** | **76.9** | 46.0 | **91.3** | 38.8 | 92.7 | 64.6 | 77.0 | 41.3 | 64.6 | 87.8 | **29.7** | 78.4 | **73.0** | 66.0 |
| SFR-Embed | 59.0 | 67.2 | 28.3 | 87.6 | 27.3 | 86.6 | 60.4 | 73.8 | 40.3 | 34.8 | **88.9** | 19.9 | 76.2 | 55.2 | 57.5 |
| Linq-Embed | 60.5 | 69.6 | 30.3 | 87.1 | 27.8 | 87.6 | 61.2 | 75.4 | 41.1 | 41.9 | 89.2 | 21.9 | 76.4 | 54.5 | 58.9 |
| e5-mistral-7b | 59.1 | 61.7 | 28.5 | 87.0 | 34.6 | 87.0 | 56.8 | 73.2 | 33.4 | 66.4 | 88.5 | 16.3 | 75.1 | 55.4 | 58.8 |
| gte-Qwen2-7B | 50.3 | 54.6 | 32.2 | 80.4 | **47.3** | 90.6 | 62.0 | 75.0 | 40.4 | 65.6 | 87.9 | 23.5 | 79.5 | 59.2 | 60.6 |
| bge-large-v1.5 | 48.7 | 64.5 | 27.3 | 74.7 | 41.9 | 87.3 | 45.0 | 75.0 | 37.1 | 51.1 | **88.9** | 22.6 | 73.6 | 47.1 | 56.1 |
| SSR-tok | 61.2 | 75.4 | 45.5 | 89.6 | 41.5 | 93.9 | 62.1 | 77.2 | 46.0 | 67.7 | 87.5 | 28.1 | 81.8 | 69.4 | 66.4 |
| SSR-CLS | 62.0 | 76.2 | **46.4** | 90.1 | 43.3 | **94.6** | **62.8** | **77.9** | **46.8** | **68.4** | 88.6 | 29.5 | **82.9** | 70.5 | **67.1** |

interactions can remain faithful to dense late interaction under suitable reconstruction and local decoder-geometry conditions.

## 4. Experiments

### 4.1. Benchmark performance

**Evaluation under Controlled Setup.** Under a controlled zero-shot setting, we compare SSR against representative state-of-the-art multi-vector dense retrievers (e.g., PLAID (Santhanam et al., 2022a)) and learned sparse retrievers (Splade-v2 (Formal et al., 2021a) and Splade-v3 (Lassance et al., 2024)). All models are trained in-domain on MSMARCO (Nguyen et al., 2016) and evaluated on both MSMARCO and 13 BEIR (Thakur et al., 2021) datasets using nDCG@10 as the primary metric. As shown in Table 1, SSR consistently delivers the best overall trade-off between effectiveness and efficiency. In particular, SSR-CLS achieves the highest average nDCG@10 of 53.4, outperforming the strongest sparse baseline Splade-v3 (51.2) and dense baseline PLAID (49.3), while SSR-tok reduces retrieval latency to 17.5ms, nearly 2× faster than ColBERTv2 and PLAID, yet still surpasses all baselines in average effectiveness. Moreover, SSR shows strong out-of-domain robustness, achieving the best results on 9 of 13 BEIR datasets. These results indicate that SSR effectively mitigates the conventional effectiveness-efficiency trade-off, and that its gains stem from the sparse coding mechanism itself rather than auxiliary token design alone. More details are in Appendix D.1.

**Evaluation on Scalability to Modern Backbone.** To examine whether SSR scales to modern large embedding backbones, we instantiate it on top of Llama-embed-nemotron-8b by freezing the backbone and training the Sparse Autoencoder on last-layer token embeddings, with the same MSMARCO training setup and sparsity constraint ($K = 32$). Table 2 compares SSR against leading open-source embed-

ding models of similar scale on the MTEB (Muennighoff et al., 2022) leaderboard. The results show that SSR can effectively unlock the rich semantic capacity of modern LLM-based encoders: SSR-CLS achieves the best average score of 67.1, outperforming strong baselines such as Qwen3-Embedding-8B (Zhang et al., 2025b) and other competitive dense retrievers including e5-mistral-7b (Wang et al., 2023). Moreover, compared with the original Llama-embed-8B backbone, our sparse adaptation yields consistent gains across tasks, with particularly clear improvements on benchmarks requiring precise entity matching, such as DBpedia and FEVER. To address the concern that SSR may benefit simply from adding an extra trained module on top of a frozen backbone, we additionally train a ColBERT-style linear projector under the same frozen-backbone setting on two representative LLM embedding backbones. This control keeps the backbone fixed and only learns a lightweight projection layer. Results in Table 3 show that ColBERT-style linear projectors improve the frozen backbones only marginally, whereas SSR achieves larger gains, indicating that the improvement comes from the sparse coding objective and retrieval mechanism rather than merely from adding an extra layer. These findings also suggest that SSR is not limited to controlled BERT-scale settings, but generalizes effectively to high-dimensional modern foundation models. More details are in Appendix D.2.

*Table 3.* **Frozen-backbone control for LLM-backbone experiments.** All methods keep the LLM backbone frozen. The linear projector controls for the effect of adding an extra trainable projection layer.

| Model | Avg. |
|---|---|
| Llama-8B | 65.8 |
| e5-mistral-7b | 58.8 |
| Llama-8B + linear | 66.1 |
| e5-mistral-7b + linear | 60.2 |
| Llama-8B + SSR-tok | 66.4 |
| Llama-8B + SSR-CLS | **67.1** |

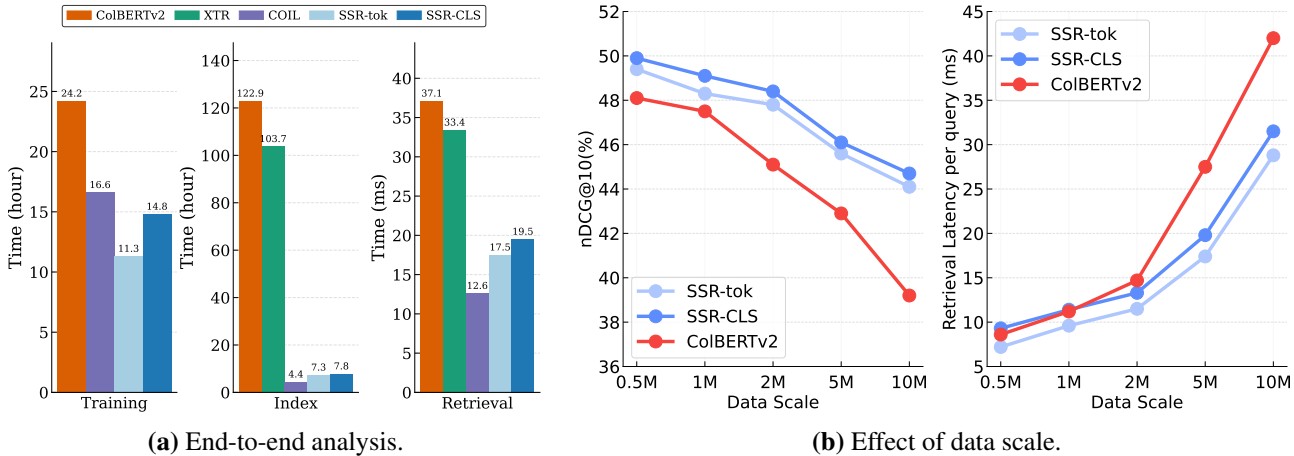

**(a)** End-to-end analysis.

**(b)** Effect of data scale.

*Figure 3.* (**Left**): Efficiency analysis on the training (left), indexing (middle) and retrieval (right) phase in the end-to-end retrieval. (**Right**): Retrieval performance (left) and efficiency (right) comparison with ColBERTv2 (Santhanam et al., 2022b) under different data scale.

**Evaluation on Robustness to Long-Tail Distributions.** We further evaluate SSR's capability to handle rare, domain-specific terminology with LoTTE (Santhanam et al., 2022b) benchmark. We find that across five corpus domains and two query sources, SSR outperforms the strongest representative baseline (i.e., ColBERTv2) with +2.5% and +2.2% on Search and Forum queries respectively. Notably, the performance gap widens in the more challenging "Pooled" setting where models must retrieve relevant documents from a massive, multi-domain merged corpus, with SSR exceeding ColBERTv2 by a substantial +3.9% on Forum queries. More detailed setup and results are available in Appendix D.3.

**Evaluation on Scalability to Long Document Sequences.** To further evaluate SSR's scalability and robustness to long sequences, we extend our evaluation to MSMARCO (Nguyen et al., 2016) document ranking subset. Compared to passage ranking set which is evaluated in Section 4.1, it presents a significantly greater challenge due to much larger document length (average sentence length from less than 100 to 1131 tokens, with many exceeding 4096). Specifically, on MS MARCO document ranking, SSR-CLS achieves the best nDCG@10 of 48.8%, while SSR-tok delivers a competitive 48.3% nDCG@10 with only 27.5ms latency. SSR achieves both performance and latency superiority compared to various dense MVR frameworks. Detailed setup and results are available in Appendix D.4.

**Stress Testing on Representational Bounds.** To stress-test the representational bound of SSR, we evaluate it on the LIMIT diagnostic benchmark (Weller et al., 2025). Unlike standard evaluation sets that measure average-case retrieval accuracy, LIMIT is designed to systematically stress-test a model's capacity to encode all possible top-$k$ document combinations within a given query space. We find that LIMIT exposes a severe representational bottleneck in stan-

dard single-vector embedding models, with Recall@5 below 5% even for state-of-the-art models such as the Qwen-Embedding series (Zhang et al., 2025b). By contrast, Multi-Vector Retrieval methods mitigate this bottleneck, and SSR establishes clear superiority, achieving 78.6% Recall@5 and 98.1% Recall@100, outperforming ColBERTv2 by 6.8 points on Recall@5. Detailed setup and results are available in Appendix D.5.

### 4.2. Efficiency and Resource Analysis

**End-to-End Efficiency Analysis.** We conduct an end-to-end efficiency analysis of the multi-vector retrieval pipeline on the MSMARCO passage ranking dataset (Nguyen et al., 2016), comparing SSR with representative state-of-the-art systems, including COIL (Gao et al., 2021), ColBERTv2 with PLAID acceleration (Santhanam et al., 2022b;a), XTR optimized by WARP (Lee et al., 2023; Scheerer et al., 2025). All models are trained and evaluated using default hyper-parameters, with the sparsity level of SSR set to $K = 32$. For a fair end-to-end comparison, the reported indexing time includes the full offline index construction pipeline after training. For SSR, this covers corpus encoding, sparse projection, inverted-list construction, list partitioning, and final block upper-bound computation. For dense MVR baselines such as ColBERTv2/PLAID and XTR/WARP, it covers corpus encoding and the final centroid-based index construction, including clustering, residual/code computation, and auxiliary retrieval structures. As shown in Figure 3(**Left**), SSR achieves substantial efficiency gains across training, indexing, and retrieval. In training, SSR-tok reduces the cost by approximately 53% compared with ColBERTv2, which requires 24.2 hours. The most significant advantage appears in indexing: while dense MVR systems such as ColBERTv2 and XTR suffer from the clustering bottleneck and require over 100 hours to process MSMARCO due to expensive K-

*Table 4.* **System-level resource breakdown.** SSR avoids clustering-based rebuilds and supports append-only index updates.

| Method | Peak mem. (GB) | Index (GB) | Update mode |
|---|---|---|---|
| ColBERTv2/PLAID | 274.2 | 22.1 | rebuild |
| XTR/WARP | 186.7 | 55.9 | rebuild |
| SSR-tok | **34.6** | **18.5** | append-only |
| SSR-CLS | 55.1 | **18.5** | append-only |

means over billions of embeddings, SSR completes indexing in about 7.5 hours, yielding over $15\times$ speedup. For online retrieval, SSR achieves sub-20ms latency, outperforming ColBERTv2 (37.1ms) and XTR (33.4ms). Although COIL is slightly faster (12.6ms), SSR offers a better balance by preserving high-precision late-interaction semantic matching while substantially improving throughput.

**CPU-Only Retrieval Efficiency.** We compare SSR's index and retrieval time compared to state-of-the-art baselines, including classic methods such as PLAID (Santhanam et al., 2022a) and recent algorithms based on ColBERT, such as IGP (Bian et al., 2025). Evaluation on MSMARCO passage ranking shows that SSR achieves superior efficiency compared to these modern MVR engines, with about 30%/85% retrieval/index time gain compared to the most efficient engine DESSERT (Engels et al., 2023). Moreover, SSR does not gain speed by sacrificing retrieval quality, resulting in the best MRR@10 among all compared methods. Experiment setup and more results are available in Appendix E.2.

**System-Level Resource Footprint.** Apart from effectiveness and efficiency, system-level resource consumption is another significant factor to evaluate MVR methods. We measure the systematic consumption along three explicit dimensions: (1) Build-time cost; (2) Serving-time footprint; (3) Maintenance/update cost. Table 4 summarizes the system-level footprint. Compared with clustering-based dense MVR engines, SSR substantially lowers peak build memory, reducing it from 274.2 GB for ColBERTv2/PLAID and 186.7 GB for XTR/WARP to 34.6 GB for SSR-tok and 55.1 GB for SSR-CLS. SSR also keeps a smaller persistent main index of 18.5 GB and supports append-only updates, since new documents only require sparse projection and posting-list insertion rather than rebuilding clustering structures. This update property is particularly important for dynamic retrieval corpora, where documents are frequently added or refreshed. Detailed experiment setups can be found in Appendix E.3.

### 4.3. Empirical Analysis

**Ablations.** We conduct ablation experiments on weights of different loss terms with sparsity level $K = 32$ during training: $\alpha$ for auxiliary loss, $\beta$ for sparse contrastive loss and $\gamma$ for supervised contrastive loss. We find that each loss term leads to performance improvement, with supervised contrastive loss resulting in the most significant performance improvement due to its promotion for understanding semantic difference between positive and negative documents. Additionally, we conduct experiments on MSMARCO passage ranking subset to isolate SSR++'s acceleration effect. Table 5 shows that SSR++ reduces the number of hit candidates from 54,278 to 3,196 and cuts retrieval latency from 38.6 ms to 17.5 ms, while maintaining essentially the same retrieval quality, with nDCG@10 changing only from 45.3 to 45.2. This shows that the coarse-to-fine sparse pruning strategy improves efficiency without introducing a meaningful effectiveness loss. More detailed results are presented in Appendix E.1.

*Table 5.* **Ablation on the SSR++'s acceleration strategy.** Evaluation is conducted on the MS MARCO passage ranking subset.

| Method | Candidates | Latency (ms) | nDCG@10 |
|---|---|---|---|
| SSR | 54278 | 38.6 | **45.3** |
| SSR++ | 3196 | **17.5** | 45.2 |

**Performance Under Different Data Scale.** To assess the effect of corpus scale on retrieval robustness and latency, we conduct a controlled MSMARCO evaluation by varying the collection size with five stratified subsets, $N \in \{0.5M, 1M, 2M, 5M, 10M\}$, randomly sampled from the full corpus. For validity, we retain only queries whose positive passages are included in each subset, and fix the sparsity level at $K = 32$ for all runs. As shown in Figure 3, larger corpora degrade performance for all models, but SSR is more robust than ColBERTv2: from 0.5M to 10M documents, SSR-tok drops by only 5.3%, compared with ColBERTv2's 8.9% decline. In terms of efficiency, SSR's inverted index-based pruning yields increasing latency advantages as $N$ grows, demonstrating its scalability for large-scale retrieval.

**Effect of Hidden Dimension $\mathbb{R}^h$.** We study how SAE's hidden dimension $h$ affects retrieval quality and latency by sweeping $h$ from $2^{12}$ to $2^{16}$, with sparsity fixed at $K = 32$ and all other hyper-parameters kept unchanged. The model is trained on MSMARCO (Nguyen et al., 2016) with a BERT-base-uncased backbone and evaluated on BEIR following Table 1. As Figure 4**(a)** shows, increasing $h$ creates a trade-off between effectiveness and efficiency. Retrieval performance follows an inverted-U trend, peaking at $h = 2^{14}$. This suggests that moderate overcompleteness provides enough capacity for the sparse codes to preserve fine-grained token interactions under the fixed Top-$K$ constraint, whereas an overly large hidden space makes the active supports more diffuse and less consistently shared across semantically related query-document pairs. This be-

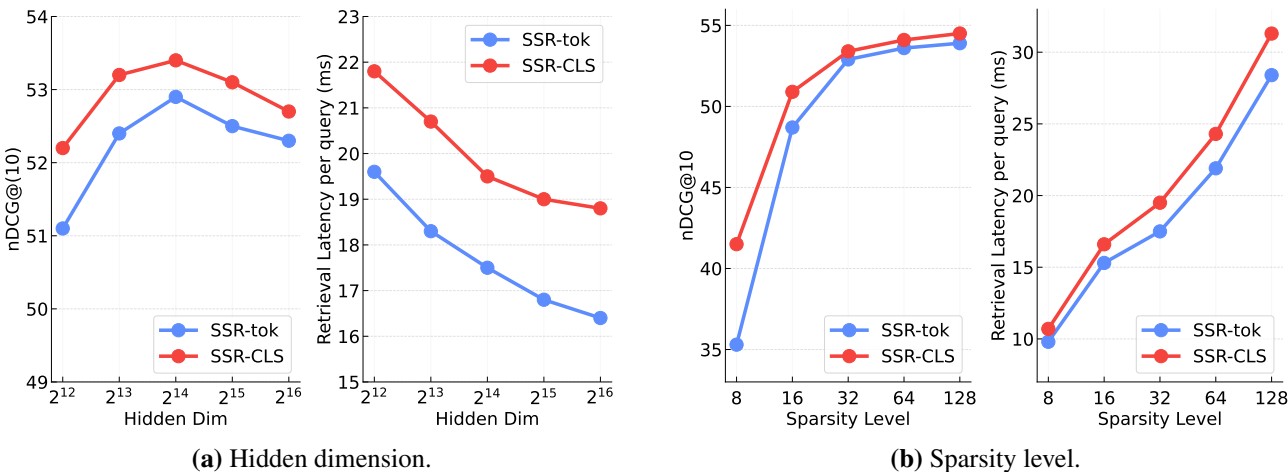

**(a)** Hidden dimension.      **(b)** Sparsity level.

*Figure 4.* **(Left)**: Effect of hidden dimension $\mathbb{R}^h$ on performance. **(Right)**: Effect of sparsity constraint $K$ on retrieval performance.

havior also highlights a key difference between sparse late interaction and dense over-parameterization. In SSR, retrieval quality depends not only on the expressiveness of individual sparse codes, but also on whether semantically related query and document tokens activate overlapping supports. When $h$ becomes too large under a fixed $K$, the activation space is fragmented: related tokens may be assigned to different neurons, reducing useful overlap in the inverted index. Therefore, the optimal hidden dimension balances feature disentanglement and support sharing.

**Sensitivity to Sparsity Constraint $K$.** We study SSR's sensitivity to the sparsity constraint $K$ by sweeping $K$ from 8 to 128, using BERT-base-uncased as the backbone with a fixed hidden dimension of $h = 16384$ and keeping all other hyperparameters unchanged. As shown in Figure 4(**b**), performance exhibits a turning point at $K = 32$: larger $K$ values bring only marginal gains, whereas reducing $K$ below 16 causes substantial degradation, likely due to the loss of fine-grained semantic information. Interestingly, the model relies more heavily on the global [CLS] token when $K$ becomes smaller, indicating that global context helps compensate for aggressively pruned token-level features. From the efficiency perspective, smaller $K$ values significantly reduce latency, mainly because they lower the cost of similarity scoring and induce a much sparser inverted index. This trade-off further motivates adaptive sparsity selection, where the model can allocate more active neurons only when queries or domains require finer-grained matching.

### 4.4. Further Discussions

**Adaptive Query-based Sparsity Control.** Since queries with different lengths may require different levels of semantic granularity, we further explore whether query sparsity can be adaptively controlled by query length. The adaptive strategy achieves comparable performance to fixed $K = 64$

(53.0% vs. 53.1%) while reducing latency from 19.9 ms to 16.3 ms, slightly improving the performance-efficiency Pareto frontier. More detailed settings and results are provided in Appendix F.1.

**Sweet Spot on Sparsity across Domains.** We also investigate whether the optimal sparsity level varies across domains. Overall, $K = 32$ is a stable choice: for example, it reaches 67.7% on fact-oriented tasks and 65.1% on multi-hop tasks. Increasing $K$ to 64 brings limited gains in most domains, but improves multi-hop retrieval from 65.1% to 66.5%. These observations suggest that sparsity in SSR should be viewed not merely as a fixed compression knob, but as a controllable interface for adapting retrieval granularity to query and domain complexity. We provide the full domain grouping and detailed comparison in Appendix F.2.

## 5. Conclusion & Discussion

We introduced *Single-stage Sparse Retrieval*, a multi-vector retrieval framework that replaces the computational bottleneck of dense clustering with efficient sparse autoencoding. By disentangling complex token semantics into high-dimensional sparse activations, SSR enables direct neuron-level inverted indexing, thereby optimizing the trade-off between indexing speed and retrieval precision. This approach is effective across both standard discriminative encoders and modern language model backbones, establishing sparse coding as a scalable alternative for next-generation retrieval systems. We believe SSR opens up a practical path toward systems that combine the semantic fidelity of multi-vector interaction with the operational simplicity of sparse inverted indexing, enabling more efficient deployment in large-scale corpora. Future work may further explore hardware-aware index layouts to improve the robustness of SSR across heterogeneous retrieval workloads.

## Impact Statement

This work aims to advance efficient large-scale retrieval in machine learning by removing the clustering bottleneck in indexing and improving empirical indexing and retrieval efficiency. These improvements may reduce the computational cost of deploying retrieval systems and make high-throughput retrieval more accessible in practical applications. However, SSR also involves important system-level trade-offs. In particular, it relies on high-dimensional sparse representations and neuron-level inverted lists, which may incur nontrivial memory and storage overhead in very large-scale deployments. The actual efficiency gains may further depend on factors such as hidden dimensionality, sparsity level, posting-list length, auxiliary index structures, hardware configuration, memory bandwidth, cache behavior, inverted-index layout, and implementation-level optimizations. Therefore, the reported latency and throughput improvements should be interpreted as empirical results under our implementation and evaluation setup, rather than hardware-independent guarantees.

However, more efficient retrieval systems may also introduce broader societal risks if deployed without appropriate safeguards. Like other high-throughput retrieval models, SSR could be used in applications that amplify existing ranking biases, accelerate the retrieval or dissemination of harmful content, or support surveillance-oriented use cases. These risks are not unique to SSR, but increased retrieval efficiency may make harmful or biased retrieval pipelines easier to scale. Responsible deployment therefore requires careful evaluation beyond standard efficiency metrics, including bias auditing, monitoring of downstream ranking behavior, content-governance mechanisms, and safeguards appropriate to the application domain.

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

## A. On the Bounded-Distortion Property of SSR

**Setup.** We provide a theoretical characterization of when Single-stage Sparse Retrieval (SSR) preserves the semantic discriminability of dense late interaction. We focus first on a pair of token embeddings, which serves as the minimal unit of late-interaction scoring, and then extend the result to the MaxSim score used in multi-vector retrieval.

Let $\mathbf{x}, \mathbf{y} \in \mathbb{R}^d$ denote two dense token embeddings. Let $\mathbf{z}_x, \mathbf{z}_y \in \mathbb{R}^h$ be their Top-$K$ sparse SAE codes. The SAE decoder reconstructs

$$\hat{\mathbf{x}} = \mathbf{W}_{\mathrm{dec}}\mathbf{z}_x + \mathbf{b}_{\mathrm{pre}}, \quad \hat{\mathbf{y}} = \mathbf{W}_{\mathrm{dec}}\mathbf{z}_y + \mathbf{b}_{\mathrm{pre}}. \tag{13}$$

Without loss of generality, we analyze the centered representation and absorb $\mathbf{b}_{\mathrm{pre}}$ into the embeddings. Thus,

$$\hat{\mathbf{x}} = \mathbf{W}_{\mathrm{dec}}\mathbf{z}_x, \quad \hat{\mathbf{y}} = \mathbf{W}_{\mathrm{dec}}\mathbf{z}_y. \tag{14}$$

For query $Q = \{\mathbf{q}_1, \ldots, \mathbf{q}_N\}$ and document $D = \{\mathbf{d}_1, \ldots, \mathbf{d}_M\}$, the dense late-interaction score is

$$S_{\mathrm{dense}}(Q, D) = \sum_{i=1}^{N} \max_{1 \le j \le M} \mathbf{q}_i^\top \mathbf{d}_j. \tag{15}$$

The corresponding sparse SSR score is

$$S_{\mathrm{SSR}}(Q, D) = \sum_{i=1}^{N} \max_{1 \le j \le M} \mathbf{z}_{q_i}^\top \mathbf{z}_{d_j}. \tag{16}$$

Since the sparse codes are Top-$K$ activated codes, the sparse inner product is equivalently computed only over overlapping activated neurons:

$$\mathbf{z}_{q_i}^\top \mathbf{z}_{d_j} = \sum_{u \in \mathcal{A}_K(\mathbf{z}_{q_i}) \cap \mathcal{A}_K(\mathbf{z}_{d_j})} \mathbf{z}_{q_i}^{(u)} \mathbf{z}_{d_j}^{(u)}. \tag{17}$$

**A sufficient regime.** The following assumptions describe a sufficient local regime under which SSR is a bounded-distortion approximation to dense late interaction.

**Assumption 1. Reconstruction fidelity.** For every dense token embedding $\mathbf{x}$ and its sparse reconstruction $\hat{\mathbf{x}} = \mathbf{W}_{\mathrm{dec}}\mathbf{z}_x$,

$$\|\mathbf{x} - \hat{\mathbf{x}}\|_2 \le \epsilon. \tag{18}$$

Equivalently, for two token embeddings $\mathbf{x}$ and $\mathbf{y}$,

$$\|\mathbf{x} - \hat{\mathbf{x}}\|_2 \le \epsilon, \quad \|\mathbf{y} - \hat{\mathbf{y}}\|_2 \le \epsilon. \tag{19}$$

**Assumption 2. Bounded dense-token norm.** Dense token embeddings are uniformly bounded:

$$\|\mathbf{x}\|_2 \le B, \quad \|\mathbf{y}\|_2 \le B. \tag{20}$$

**Assumption 3. Restricted decoder near-orthogonality.** Let

$$S = \mathrm{supp}(\mathbf{z}_x) \cup \mathrm{supp}(\mathbf{z}_y) \tag{21}$$

be the union of active sparse coordinates. The decoder is $\delta$-near-orthogonal on this active support:

$$\left\| \mathbf{W}_{\mathrm{dec}}^\top \mathbf{W}_{\mathrm{dec}} - \mathbf{I} \right\|_{S \to S} \le \delta. \tag{22}$$

That is, for all sparse vectors $\mathbf{a}$ and $\mathbf{b}$ supported on $S$,

$$\left| \mathbf{a}^\top \left( \mathbf{W}_{\mathrm{dec}}^\top \mathbf{W}_{\mathrm{dec}} - \mathbf{I} \right) \mathbf{b} \right| \le \delta \|\mathbf{a}\|_2 \|\mathbf{b}\|_2. \tag{23}$$

**Assumption 4. Bounded sparse-code norm.** When extending the result to full query-document scoring, we assume the sparse codes are uniformly bounded:

$$\|\mathbf{z}_x\|_2 \le C, \quad \|\mathbf{z}_y\|_2 \le C. \tag{24}$$

This assumption is used only to obtain a uniform late-interaction bound.

**Theorem A** (Token-level bounded distortion). Under **Assumptions 1–3**, the dense token inner product and the SSR sparse inner product satisfy

$$\left|\mathbf{x}^\top\mathbf{y} - \mathbf{z}_x^\top\mathbf{z}_y\right| \le 2B\epsilon + \epsilon^2 + \delta\|\mathbf{z}_x\|_2\|\mathbf{z}_y\|_2. \tag{25}$$

If **Assumption 4** also holds, then

$$\left|\mathbf{x}^\top\mathbf{y} - \mathbf{z}_x^\top\mathbf{z}_y\right| \le 2B\epsilon + \epsilon^2 + \delta C^2. \tag{26}$$

**Proof.** We decompose the approximation error into a reconstruction term and a decoder-geometry term:

$$\left|\mathbf{x}^\top\mathbf{y} - \mathbf{z}_x^\top\mathbf{z}_y\right| \le \left|\mathbf{x}^\top\mathbf{y} - \hat{\mathbf{x}}^\top\hat{\mathbf{y}}\right| + \left|\hat{\mathbf{x}}^\top\hat{\mathbf{y}} - \mathbf{z}_x^\top\mathbf{z}_y\right|. \tag{27}$$

We first bound the reconstruction term. Let

$$\mathbf{e}_x = \mathbf{x} - \hat{\mathbf{x}}, \quad \mathbf{e}_y = \mathbf{y} - \hat{\mathbf{y}}. \tag{28}$$

Then $\hat{\mathbf{x}} = \mathbf{x} - \mathbf{e}_x$ and $\hat{\mathbf{y}} = \mathbf{y} - \mathbf{e}_y$. Therefore,

$$\begin{aligned}
\mathbf{x}^\top\mathbf{y} - \hat{\mathbf{x}}^\top\hat{\mathbf{y}} &= \mathbf{x}^\top\mathbf{y} - (\mathbf{x} - \mathbf{e}_x)^\top(\mathbf{y} - \mathbf{e}_y) \\
&= \mathbf{x}^\top\mathbf{e}_y + \mathbf{e}_x^\top\mathbf{y} - \mathbf{e}_x^\top\mathbf{e}_y.
\end{aligned} \tag{29}$$

By Cauchy-Schwarz,

$$\left|\mathbf{x}^\top\mathbf{y} - \hat{\mathbf{x}}^\top\hat{\mathbf{y}}\right| \le \|\mathbf{x}\|_2\|\mathbf{e}_y\|_2 + \|\mathbf{e}_x\|_2\|\mathbf{y}\|_2 + \|\mathbf{e}_x\|_2\|\mathbf{e}_y\|_2. \tag{30}$$

Using **Assumptions 1–2**, we obtain

$$\left|\mathbf{x}^\top\mathbf{y} - \hat{\mathbf{x}}^\top\hat{\mathbf{y}}\right| \le 2B\epsilon + \epsilon^2. \tag{31}$$

We now bound the decoder-geometry term. Since

$$\hat{\mathbf{x}} = \mathbf{W}_{\text{dec}}\mathbf{z}_x, \quad \hat{\mathbf{y}} = \mathbf{W}_{\text{dec}}\mathbf{z}_y, \tag{32}$$

we have

$$\hat{\mathbf{x}}^\top\hat{\mathbf{y}} = \mathbf{z}_x^\top\mathbf{W}_{\text{dec}}^\top\mathbf{W}_{\text{dec}}\mathbf{z}_y. \tag{33}$$

Thus,

$$\hat{\mathbf{x}}^\top\hat{\mathbf{y}} - \mathbf{z}_x^\top\mathbf{z}_y = \mathbf{z}_x^\top\left(\mathbf{W}_{\text{dec}}^\top\mathbf{W}_{\text{dec}} - \mathbf{I}\right)\mathbf{z}_y. \tag{34}$$

By **Assumption 3**,

$$\left|\hat{\mathbf{x}}^\top\hat{\mathbf{y}} - \mathbf{z}_x^\top\mathbf{z}_y\right| \le \delta\|\mathbf{z}_x\|_2\|\mathbf{z}_y\|_2. \tag{35}$$

Combining the two bounds gives

$$\left|\mathbf{x}^\top\mathbf{y} - \mathbf{z}_x^\top\mathbf{z}_y\right| \le 2B\epsilon + \epsilon^2 + \delta\|\mathbf{z}_x\|_2\|\mathbf{z}_y\|_2. \tag{36}$$

If $\|\mathbf{z}_x\|_2, \|\mathbf{z}_y\|_2 \le C$, then

$$\left|\mathbf{x}^\top\mathbf{y} - \mathbf{z}_x^\top\mathbf{z}_y\right| \le 2B\epsilon + \epsilon^2 + \delta C^2. \tag{37}$$

This completes the proof. □

**Remark.** Theorem A shows that, at the token level, SSR is a bounded-distortion approximation to dense interaction. The distortion consists of two parts. The first part,

$$2B\epsilon + \epsilon^2, \tag{38}$$

comes from SAE reconstruction error. The second part,

$$\delta\|\mathbf{z}_x\|_2\|\mathbf{z}_y\|_2, \tag{39}$$

comes from the deviation of the decoder geometry from an orthonormal embedding of the active sparse coordinates. Therefore, if dense token embeddings are well reconstructed and the decoder is approximately orthogonal on the active support, then

$$\mathbf{x}^\top\mathbf{y} = \mathbf{z}_x^\top\mathbf{z}_y + O(B\epsilon + \epsilon^2) + O(\delta\|\mathbf{z}_x\|_2\|\mathbf{z}_y\|_2). \tag{40}$$

In particular, under bounded sparse-code norms,

$$\mathbf{x}^\top \mathbf{y} = \mathbf{z}_x^\top \mathbf{z}_y + O(B\epsilon + \epsilon^2 + \delta C^2). \tag{41}$$

**Theorem B** (Late-interaction bounded distortion). Suppose **Assumptions 1–4** hold uniformly for every query-document token pair $(\mathbf{q}_i, \mathbf{d}_j)$. Define

$$\eta := 2B\epsilon + \epsilon^2 + \delta C^2. \tag{42}$$

Then the dense late-interaction score and the SSR sparse late-interaction score satisfy

$$|S_{\text{dense}}(Q, D) - S_{\text{SSR}}(Q, D)| \leq N\eta. \tag{43}$$

**Proof.** Let

$$s_{ij} = \mathbf{q}_i^\top \mathbf{d}_j, \quad \tilde{s}_{ij} = \mathbf{z}_{q_i}^\top \mathbf{z}_{d_j}. \tag{44}$$

By Theorem A and the uniform boundedness assumptions,

$$|s_{ij} - \tilde{s}_{ij}| \leq \eta, \quad \forall i, j. \tag{45}$$

For each query token embedding $\mathbf{q}_i$, we use the elementary inequality

$$\left| \max_j s_{ij} - \max_j \tilde{s}_{ij} \right| \leq \max_j |s_{ij} - \tilde{s}_{ij}| \leq \eta. \tag{46}$$

Therefore,

$$\begin{aligned}
|S_{\text{dense}}(Q, D) - S_{\text{SSR}}(Q, D)| &= \left| \sum_{i=1}^N \max_j s_{ij} - \sum_{i=1}^N \max_j \tilde{s}_{ij} \right| \\
&\leq \sum_{i=1}^N \left| \max_j s_{ij} - \max_j \tilde{s}_{ij} \right| \\
&\leq N\eta.
\end{aligned} \tag{47}$$

This completes the proof. $\square$

**Consequence.** Combining Theorem A and Theorem B, SSR preserves dense late-interaction scores up to a bounded distortion. At the token level, if dense token embeddings $\mathbf{x}, \mathbf{y}$ are well reconstructed, i.e.,

$$\|\mathbf{x} - \hat{\mathbf{x}}\|_2, \ \|\mathbf{y} - \hat{\mathbf{y}}\|_2 < \epsilon, \tag{48}$$

and their norms are bounded as

$$\|\mathbf{x}\|_2, \ \|\mathbf{y}\|_2 < B, \tag{49}$$

then the dense similarity differs from the reconstructed dense similarity by at most

$$O(B\epsilon + \epsilon^2). \tag{50}$$

Moreover, if the decoder is approximately orthogonal on the active support, namely

$$\left\| \mathbf{W}_{\text{dec}}^\top \mathbf{W}_{\text{dec}} - \mathbf{I} \right\|_{S \to S} \leq \delta, \tag{51}$$

then the reconstructed dense similarity is further close to the sparse inner product:

$$\hat{\mathbf{x}}^\top \hat{\mathbf{y}} = \mathbf{z}_x^\top \mathbf{z}_y + O(\delta \|\mathbf{z}_x\|_2 \|\mathbf{z}_y\|_2). \tag{52}$$

Therefore,

$$\mathbf{x}^\top \mathbf{y} = \mathbf{z}_x^\top \mathbf{z}_y + O(B\epsilon + \epsilon^2) + O(\delta \|\mathbf{z}_x\|_2 \|\mathbf{z}_y\|_2). \tag{53}$$

Equivalently, SSR is a bounded-distortion approximation to dense late interaction:

$$\left| \mathbf{x}^\top \mathbf{y} - \mathbf{z}_x^\top \mathbf{z}_y \right| = O(B\epsilon + \epsilon^2) + O(\delta \|\mathbf{z}_x\|_2 \|\mathbf{z}_y\|_2). \tag{54}$$

For the full MaxSim late-interaction score, if the above conditions hold uniformly over all query-document token pairs and $\|\mathbf{z}_{q_i}\|_2, \|\mathbf{z}_{d_j}\|_2 \leq C$, then

$$|S_{\text{dense}}(Q, D) - S_{\text{SSR}}(Q, D)| \leq N \left(2B\epsilon + \epsilon^2 + \delta C^2\right). \tag{55}$$

Thus, when the SAE reconstruction error is small and the decoder is near-orthogonal on active sparse supports, SSR preserves the semantic discriminability of dense late interaction while replacing dense token similarity with sparse overlap-based scoring.

# B. Additional Related Work

**Single-Vector Retrieval in the LLM Era.** Leveraging the generative capabilities of Large Language Models (LLMs), recent research has significantly advanced single vector representation for documents (Liu et al., 2021; 2022; Cao et al., 2023; Zhou et al., 2023; You et al., 2024; Zhang et al., 2024; Min et al., 2025; You et al., 2025; Xie et al., 2026; Wei et al., 2026). Through techniques such as synthetic data generation (Sturua et al., 2024; Lee et al., 2024), hard negative mining (Zhang et al., 2025b; Wang et al., 2023), and instruction tuning (Peng et al., 2024), representative models like Llama-Embed-8B (Babakhin et al., 2025) and Kalm-embedding-v2 (Zhao et al., 2025) have achieved state-of-the-art performance on the MTEB benchmark (Muennighoff et al., 2022). To mitigate the computational and storage cost, various compression techniques have been proposed. MRL (Kusupati et al., 2022) employs multi-length representation learning, allowing for adaptive truncation with minimal performance loss. CSR (Wen et al., 2025; Guo et al., 2026) and the Splade series (Formal et al., 2021b;a; Lassance et al., 2024) design sparse structure for high-efficiency retrieval, respectively exploring sparse projections and lexical expansions.

**Multi-Vector Retrieval.** Following the fine-grained interaction paradigm established by ColBERT (Khattab & Zaharia, 2020), recent research has focused on addressing its scalability bottlenecks. On the storage front, ColBERTv2 (Santhanam et al., 2022b), CRISP (Veneroso et al., 2025), and EMVB (Nardini et al., 2024) significantly reduce memory footprints through techniques ranging from residual compression to product quantization. To enhance retrieval efficiency, COIL (Gao et al., 2021) and CITADEL (Li et al., 2023a) adopt inverted list structures for faster indexing, while PLAID (Santhanam et al., 2022a) and EMVB (Nardini et al., 2024) implement multi-stage pruning mechanisms. More recent works optimize the entire pipeline semantics: XTR (Lee et al., 2023) improves candidate quality via a token-retrieval objective, and WARP (Scheerer et al., 2025) further optimizes this with dynamic similarity imputation and implicit decompression. Finally, ALIGNER (Qian et al., 2022) and LITE (Ji et al., 2024) explore alternative interaction optimizations via salience pruning and learnable MLP scoring, respectively. There have also been a few recent works that specially optimize the index and retrieval of ColBert-style models, such as IGP (Bian et al., 2025), MUVERA (Dhulipala et al., 2024) and DESSERT (Engels et al., 2023).

**Sparse Autoencoder(SAE).** Sparse coding algorithms (Lee et al., 2006) serve as powerful techniques for compressing high-dimensional features into semantically meaningful, sparse representations, among which Sparse Autoencoder (SAE) pioneers for adaptive representation (Wen et al., 2025; Guo et al., 2026) and foundation model interpretability (Zhang et al., 2025a; Cunningham et al., 2023; Templeton et al., 2024). SAEs project these high-dimensional vectors into sparse feature directions, effectively translating abstract model activations into human-readable concept dictionaries. To enhance the precision of this semantic extraction, recent studies have introduced advanced sparsity control mechanisms, ranging from TopK and Batch TopK constraints (Gao et al., 2024; Bussmann et al., 2024) to dynamic activation functions like JumpReLU and Gated architectures (Rajamanoharan et al., 2024b;a).

## C. Datasets

Three distinct categories of datasets are involved in this paper: the BEIR benchmark (Thakur et al., 2021) for diverse retrieval tasks, LoTTE (Santhanam et al., 2022b) for long-tail retrieval, and LIMIT (Weller et al., 2025) for rigorous stress testing.

- **Arguana(AR) (Wachsmuth et al., 2018)**: Arguana comprises 6753 argument-counterargument pairs extracted from 1069 debates on idebate.org across 15 diverse themes, uniquely featuring explicit counterarguments that attack the premises or conclusions of specific points.

- **ClimateFever(CF) (Diggelmann et al., 2020)**: ClimateFever is a challenging real-world dataset for climate-related claim verification, consisting of 1,535 claims collected from the internet and paired with 7,675 Wikipedia-sourced evidence sentences annotated as supporting, refuting, or providing insufficient information.

- **DBpedia(DB) (Hasibi et al., 2017)**: DBpedia is a standard and updated test collection for entity search that provides graded relevance judgments for 467 queries across various categories, mapped to 4.6 million entities from DBpedia.

- **Fever(FE) (Thorne et al., 2018)**: FEVER is a large-scale dataset consisting of 185,445 claims derived from Wikipedia, where systems are tasked with retrieving sentence-level evidence to classify each claim as Supported, Refuted, or NotEnoughInfo.

- **FiQA-2018(FQ) (Maia et al., 2018)**: Constructed from the StackExchange Investment community, the FiQA-2018 QA dataset focuses on opinion-based question answering over financial data, containing a knowledge base of 57,640 posts and over 17,000 labeled question-answer pairs for training.

- **HotpotQA(HQ) (Yang et al., 2018)**: HotpotQA is a large-scale dataset containing about 113k Wikipedia-based question-answer pairs designed to test multi-hop reasoning, with sentence-level supporting facts provided for explainability.

- **MSMARCO(MS) (Nguyen et al., 2016)**: MS MARCO is a large-scale machine reading comprehension dataset comprising real-world user queries sampled from Bing search logs, where answers are human-generated summaries derived from retrieved web documents rather than simple extracted spans.

- **NFCorpus(NF) (Boteva et al., 2016)**: NFCorpus is a full-text learning-to-rank dataset in the medical domain, consisting of over 3,000 layman's queries collected from NutritionFacts.org mapped to technical medical abstracts from PubMed with three-level relevance judgments.

- **NQ (Kwiatkowski et al., 2019)**: Natural Questions (NQ) is a large-scale dataset consisting of 307,373 real-world Google search queries paired with Wikipedia pages, where systems are tasked with identifying a long answer (e.g., a paragraph) and a short answer (e.g., an entity), or determining if no answer exists.

- **Quora(QU) (Thakur et al., 2021)**: Quora is a duplicate question retrieval dataset where systems are tasked with retrieving semantically equivalent questions from a large corpus of community-generated questions given an input query.

- **SCIDOCS(SD) (Cohan et al., 2020)**: SCIDOCS is a scientific document evaluation benchmark where the retrieval task involves predicting citation links between research papers based on their titles and abstracts.

- **SciFact(SF) (Wadden et al., 2020)**: SciFact is a scientific fact-checking dataset where the retrieval task involves identifying research abstracts containing evidence that either supports or refutes a given expert-written claim.

- **TREC-COVID(CV) (Voorhees et al., 2021)**: TREC-COVID is an ad-hoc retrieval dataset constructed from the CORD-19 corpus, consisting of scientific articles related to COVID-19 and coronaviruses paired with pandemic-related queries annotated by biomedical experts.

- **Touché-2020(TO) (Bondarenko et al., 2020)**: Focusing on argument retrieval, Touché-2020 consists of search scenarios on controversial issues where the goal is to retrieve grounded arguments that comprise conclusions and premises from online debate portals.

- **LoTTE** (Santhanam et al., 2022b): LoTTE evaluates out-of-domain retrieval performance on information-seeking queries across long-tail topics. It encompasses five domains, including writing, recreation, science, technology, and lifestyle, along with an aggregated pooled setting. To capture varying user intents, each domain provides two distinct query subsets: search queries sourced from GooAQ and forum queries derived from StackExchange post titles.

- **LIMIT** (Weller et al., 2025): Designed as a diagnostic retrieval benchmark, LIMIT empirically probes the theoretical representational bounds of embedding models. Unlike standard evaluation sets, it systematically stress-tests a model's capacity to encode all possible top-$k$ document combinations within a given query space.

# D. Details on Benchmark Performance Experiments

## D.1. Evaluation under Controlled Setup

**Baselines.** We benchmark against two categories of state-of-the-art retrieval models:

- **Multi-vector dense retrievers**: We include late-interaction models such as ColBERT (Khattab & Zaharia, 2020) and its efficiency-optimized variants ColBERTv2 (Santhanam et al., 2022b) and PLAID (Santhanam et al., 2022a)). Additionally, we compare against recent state-of-the-art advancements, like AligneR (Qian et al., 2022), CITADEL (Li et al., 2023a), and XTR (Lee et al., 2023). Moreover, COIL (Gao et al., 2021), which utilizes inverted index structure, is analyzed.

- **Sparse retrievers**: We select the Splade family (Splade-v2 (Formal et al., 2021a) and Splade-v3 (Lassance et al., 2024)) as the primary sparse baseline, which currently represents the state-of-the-art in learned sparse retrieval.

**Experiment Setup.** We conduct evaluations under a rigorous zero-shot setting to assess generalization capabilities. All models are trained in-domain on the MSMARCO (Nguyen et al., 2016) passage ranking dataset with negative documents sampled by BM25 (Robertson et al., 1995) and evaluated on both in-domain MSMARCO and 13 out-of-domain datasets from the BEIR (Thakur et al., 2021) benchmark. where primary evaluation metric is nDCG@10. Intuitively, nDCG@10 computes a weighted sum of relevance scores for the top 10 result and is normalized against the maximum possible score. To ensure a fair comparison, SSR is trained on MSMARCO using BERT-base-uncased (Devlin et al., 2019) backbone, with sparsity level $K$ fixed at 32 and max sequence length set 32. Both SSR and SSR+CLS have been accelerated by SSR++ pruning. Latency is measured on A100 40G GPU, after 1000 warmup queries.

**Implementation Details.** During training, we include two separate SAEs: one for regular token embedding ($E_{\text{tok}}$ and one for the global [CLS] token ($E_{\text{tok}}$). Setup for hyper-parameters in Table 6. During evaluation, SSR++ partitions lists into blocks of size 64, uses the top-4 activated neurons for coarse pruning, keeps top-2000 candidates and then applies exact refinement with full $K = 32$.

*Table 6.* Implementation details on evaluation under controlled setup.

| d | h | Top$K$ | $K_{\text{coarse}}$ | Block Size | $k_{\text{aux}}$ | Optimizer | lr | Batch Size | Epoch | Warmup | $\gamma$ | $\alpha$ | $\beta$ |
|---|---|---|---|---|---|---|---|---|---|---|---|---|---|
| 768 | 16384 | 32 | 4 | 64 | 2048 | AdamW | 0.001 | 64 | 2 | 20000 | 0.05 | 0.03125 | 0.1 |

## D.2. Evaluation on Scalability to Modern Backbone

**Baselines.** We choose several embedding models that are competitive on MTEB benchmark (Muennighoff et al., 2022). These models are Qwen3-Embedding-8B (Zhang et al., 2025b), SFR-Embedding-Mistral (Rui Meng et al., 2024), Linq-Embed-Mistral (Choi et al., 2024), e5-mistral-7b-instruct (Wang et al., 2023), gte-Qwen2-7B-instruct (Li et al., 2023b) and bge-large-en-v1.5 (Xiao et al., 2023).

**Experiment Setup.** We use Llama-embed-nemotron-8b as backbone. During training, we freeze the backbone parameters and train Sparse Autoencoder directly on the last-layer token embeddings. Training is conducted on the MSMARCO passage ranking dataset with the sparsity constraint $K$ maintained at 32. Detailed setup is presented in Table 7, with other parameters set as the same in Table 6.

*Table 7.* Implementation details on evaluation on scalability to modern backbone.

| d | h | Precision | lr | Global Batch Size |
|---|---|---|---|---|
| 4096 | 65536 | BF16 | 0.0002 | 32 |

## D.3. Evaluation on Robustness to Long-Tail Distribution.

**Datasets and Baselines.** We benchmark SSR on the LoTTE (Santhanam et al., 2022b) dataset with five specialized domains (writing, recreation, science, technology, and lifestyle) and an aggregated pooled setting. Each domain includes

two query sources, with search queries sourced from GooAQ and forum queries derived from StackExchange post titles. We compare against two representative MVR frameworks, XTR and ColBERTv2.

**Experimental Setup**  We utilize the same training and evaluation configurations as described in Appendix D.2 to ensure a fair comparison, except that the max sequence length is set 1024 rather than 512.

**Evaluation Results**  Table 8 demonstrates SSR's performance on LOTTE (Santhanam et al., 2022b). Results show that SSR achieves superior performance compared with representative baselines.

*Table 8.* **Evaluation on Long-Tail Benchmark LoTTE (Santhanam et al., 2022b).** We report Success@5 as evaluation metrics, where the maximum values are indicated in **bold**.

| Query Type | Model | Corpus | | | | | | |
|---|---|---|---|---|---|---|---|---|
| | | **Writing** | **Recreation** | **Science** | **Technology** | **Lifestyle** | **Pooled** | **Avg.** |
| Search | ColBERTv2 | 79.2 | 70.8 | 55.5 | 65.9 | 84.3 | 71.8 | 71.3 |
| | XTR | 78.6 | 69.5 | 56.1 | 64.6 | 83.5 | 69.3 | 70.3 |
| | SSR | **81.8** | **73.9** | **59.3** | **67.1** | **87.6** | **73.3** | **73.8** |
| Forum | ColBERTv2 | 78.7 | 71.4 | 45.8 | 53.9 | 77.3 | 63.5 | 65.1 |
| | XTR | 76.4 | 70.9 | 44.2 | 53.4 | 75.8 | 61.7 | 63.6 |
| | SSR | **81.8** | **74.3** | **47.3** | **55.9** | **79.7** | **67.4** | **67.3** |

## D.4. Evaluation on Scalability to Long Document Sequences

**Datasets and Baselines.**  We compare against a few representative baselines, including ColBERTv2 (Santhanam et al., 2022b), XTR (Lee et al., 2023) and COIL (Gao et al., 2021). Both MSMARCO passage and document ranking subsets are evaluated to demonstrate performance and efficiency difference on various document lengths in similar domain.

**Experiment Setup.**  We utilize the same training and evaluation configurations as described in Appendix D.2, except that the max sequence length is set 4096 rather than 512. Similarity, SSR has been accelerated by SSR++.

*Table 9.* **Performance and efficiency comparison on Long-Context Retrieval.** We report nDCG@10 (%) for retrieval quality and retrieval time per query (ms) for efficiency. The maximum values are indicated in **bold**.

| Method | MS Passage | | MS Document | |
|---|---|---|---|---|
| | nDCG@10 (%) | latency (ms) | nDCG@10 (%) | latency (ms) |
| ColBERTv2 | 39.8 | 37.1 | 41.5 | 79.3 |
| XTR | **47.6** | 45.0 | 48.1 | 52.6 |
| COIL | 35.3 | 12.6 | 38.8 | 18.8 |
| SSR-tok | 45.2 | 17.5 | 48.3 | 27.5 |
| SSR-CLS | 45.5 | 19.5 | **48.8** | 29.3 |

**Evaluation Results.**  Results in Table 9 demonstrate that while dense interaction mechanism of ColBERTv2 causes latency to spike to 79.3ms on longer documents, SSR-CLS maintains a sub-30ms latency (29.3ms), achieving a 2.7× speedup by leveraging sparse inverted indexing to restrict interactions to activated neurons. Crucially, this efficiency gain incurs no performance penalty: SSR-CLS attains the highest retrieval accuracy (48.8), outperforming the long-context optimized XTR (48.1) and significantly surpassing the lexical-only COIL baseline, thereby confirming its capability to preserve fine-grained details over long sequences with high throughput.

## D.5. Stress Testing on Representational Bounds.

**Baselines and Experiment Setup.**  We select three outstanding SVR models (Qwen3-Embedding-4B, GritLM-7B and e5-mistral-7B-instruct) and two MVR frameworks (ColBERTv2 and XTR) to fully compare the representational bounds across different methods and paradigms. Same training configuration as described in Appendix D.1 is utilized, with sparsity $K = 32$.

**Evaluation Results.**  As shown in Table 10, the LIMIT benchmark exposes a severe representational bottleneck in standard

embedding models, with Single-Vector Retrieval (SVR) systems experiencing catastrophic failure (e.g., scoring below 5% on Recall@5). This empirically validates that forcing diverse document semantics into a single fixed-length vector causes irreversible information loss under stress. While dense Multi-Vector Retrieval (MVR) methods like ColBERTv2 mitigate this bottleneck by preserving token-level granularity, SSR establishes clear superiority by achieving a Recall@5 of 78.6% and a Recall@100 of 98.1%. SSR outperforms the strongest dense baseline (ColBERTv2) by substantial margins (+6.8% on Recall@5).

*Table 10.* **Stress Testing on various SVR and MVR methods.** We report Recall@5, Recall@10 and Recall@100 as evaluation metrics, where the maximum values are indicated in **bold**.

| Method | Recall@5 | Recall@10 | Recall@100 |
|---|---|---|---|
| **Single-Vector Retrieval** | | | |
| Qwen3-4B | 1.2 | 4.6 | 6.3 |
| GritLM 7B | 4.7 | 6.1 | 19.4 |
| e5-mistral-7B | 2.8 | 5.2 | 10.9 |
| **Multi-Vector Retrieval** | | | |
| ColBERTv2 | 71.8 | 82.6 | 94.4 |
| XTR | 68.4 | 79.9 | 93.2 |
| SSR | **78.6** | **87.3** | **98.1** |

# E. Details on Efficiency and Empirical Analysis

## E.1. Ablations

Table 11, 12 and 13 respectively demonstrates different settings on loss weights $\alpha$, $\beta$ and $\gamma$, with other parameters set as default as Appendix D.1. Evaluation is done on BEIR benchmark.

*Table 11.* Different settings on auxiliary loss weight $\alpha$.

| $\alpha$ | 1/64 | 1/32 | 1/16 | 0.1 | 0.2 |
|---|---|---|---|---|---|
| SSR-tok | 52.1 | **52.9** | 52.4 | 51.6 | 50.8 |
| SSR-CLS | 52.3 | **53.4** | 52.7 | 51.8 | 51.3 |

*Table 12.* Different settings on sparse contrastive loss weight $\beta$.

| $\beta$ | 0.05 | 0.1 | 0.15 | 0.2 | 0.25 |
|---|---|---|---|---|---|
| SSR-tok | 52.5 | **52.9** | 52.8 | 52.3 | 51.5 |
| SSR-CLS | 53.1 | **53.4** | 53.1 | 52.7 | 51.6 |

*Table 13.* Different settings on supervised contrastive loss weight $\gamma$.

| $\gamma$ | 0.01 | 0.02 | 0.05 | 0.1 | 0.2 |
|---|---|---|---|---|---|
| SSR-tok | 49.7 | 51.3 | **52.9** | 50.4 | 47.6 |
| SSR-CLS | 50.3 | 51.7 | **53.4** | 51.2 | 48.5 |

We further conduct experiments to ablate each loss term respectively. Results in Table 14 show that each loss term leads to performance improvement while supervised contrastive loss results in the most significant improvement. Sparsity level $K$ is set 32.

*Table 14.* Ablation on each loss term.

| Loss term | | | SSR-tok | SSR-CLS |
|---|---|---|---|---|
| $\alpha$ | $\beta$ | $\gamma$ | | |
| 0 | 0 | 0 | 46.4 | 46.7 |
| 1/32 | 0 | 0 | 47.5 | 48.6 |
| 1/32 | 0.1 | 0 | 48.2 | 49.5 |
| 1/32 | 0.1 | 0.05 | 52.9 | 53.4 |

## E.2. CPU-based Efficiency Analysis

**Baselines and Experiment Setup.** We compare against six MVR engines, including PLAID (Santhanam et al., 2022a), WARP (Scheerer et al., 2025), EMVB (Nardini et al., 2024), DESSERT (Engels et al., 2023), MUVERA (Dhulipala et al., 2024) and IGP (Bian et al., 2025). Index and Retrieval are done on Intel(R) Xeon(R) Platinum 8275CL CPU @ 3.00GHz with 96 cores. Retrieval depth is set 100 in retrieval efficiency calculation, while other settings are set as the same in the original papers.

**Results.** Table 15 demonstrates different methods' performance (MRR@10), index time (hour) and retrieval time (ms). Results show that SSR achieve better performance-efficiency trade-off compared to various modern engines.

## E.3. System Resource Analysis

**Baselines and Experiment Setup.** We compare SSR against two representative engines: ColBERTv2 (accelerated by PLAID (Santhanam et al., 2022a)) and XTR (accelerated by WARP (Scheerer et al., 2025)). We analyze through three dimensions:

*Table 15.* CPU-based Efficiency Analysis (the best results are highlighted in **bold**).

| Method | MRR@10(%) | Index(h) | Retrieval(ms) |
|---|---|---|---|
| PLAID | 39.6 | 122.9 | 156.3 |
| WARP | 38.7 | 103.7 | 129.7 |
| EMVB | 39.8 | 61.5 | 93.4 |
| DESSERT | 37.2 | 49.8 | 75.6 |
| MUVERA | 39.7 | 82.6 | 94.7 |
| IGP | 39.1 | 119.7 | 113.2 |
| SSR-tok | 39.5 | 7.3 | **49.1** |
| SSR-CLS | **40.2** | **7.8** | 57.3 |

- **Build-time cost:** the peak build-time memory during training and indexing

- **Serving-time footprint:** the persistent main index size from transient build resources, including auxiliary retrieval structures such as centroids/codebooks/residual metadata for PLAID-style systems, and posting lists/block metadata for SSR.

- **Maintenance/update cost:** the update mode when new candidate documents appear after initial indexing.

# F. Further Discussions

## F.1. Adaptive Query-based Sparsity Control

The optimal sparsity level for different queries may be different as longer queries may contain more complex semantic information, therefore requiring more fine-grained features for effective retrieval. To fully explore the performance-efficiency trade-off across different sparsity settings, we evaluate on BEIR in four query sparsity settings with models trained in benchmark experiments under controlled setup (i.e., Section 4.1): fixed sparsity $K = 16$, $K = 32$, $K = 64$ and adaptive sparsity based on query length. To be specific, for query with less than 3 tokens, $K$ is set 16, while 32 for those with 4-7 tokens and 64 for those with more than 8 tokens. Table 16 shows that adaptive sparsity slightly improves the Pareto frontier of performance-efficiency in MVR compared to fixed strategies.

Table 16. Comparison of sparsity settings based on query length.

| Sparsity Setting | Performance(%) | Latency(ms) |
|:---:|:---:|:---:|
| Fixed-16 | 50.5 | 16.4 |
| Fixed-32 | 52.9 | 17.5 |
| Fixed-64 | 53.1 | 19.9 |
| Adaptive | 53.0 | 16.3 |

## F.2. Sweet Spot on Sparsity across Domains

Due to difference in semantic complexity, the most stable sparsity setting may vary for different domains. We select 11 representative tasks in BEIR, divide them into 4 domains based on content and analyze each domain's average performance under different $K$:

- **fact**: DBpedia, Fever, SciFact

- **multi-hop**: NQ, HotpotQA

- **scientific**: TREC-COVID, NFCorpus, SCIDOCS

- **opinion**: Arguana, Touché-2020, FiQA-2018

Results in Table 17 show that $K = 32$ is the most stable setting for most domains. However, for domains whose semantics tend to be clear and easily separable (**fact**), $K = 16$ leads to relatively minor performance degradation while for domains where more fine-grained search is needed (**multi-hop**), $K = 64$ results in obvious performance improvement but introduces additional latency cost.

Table 17. Sparsity Setting Difference across domains.

| K | 8 | 16 | 32 | 64 | 128 |
|:---:|:---:|:---:|:---:|:---:|:---:|
| **fact** | 50.4 | 65.9 | 67.7 | 68.2 | 68.4 |
| **multi-hop** | 50.2 | 58.6 | 65.1 | 66.5 | 66.8 |
| **scientific** | 29.3 | 39.6 | 44.2 | 44.8 | 45.3 |
| **opinion** | 20.9 | 33.3 | 39.5 | 40.3 | 40.6 |

