# OpenReview forum: "No More K-means: Single-Stage Sparse Coding for Efficient Multi-Vector Retrieval"
_ICML.cc/2026/Conference — ICML 2026 regular_

### Official Review · Reviewer_m84N · 2026-02-20

**Soundness:** 2
**Presentation:** 1
**Significance:** 3
**Originality:** 3
**Overall Recommendation:** 5
**Confidence:** 4

**Summary:**

The problem the authors consider is information retrieval, i.e., finding the relevant documents for a query from a large corpus. Specifically, the authors propose a multi-vector retrieval (MVR) method. In multi-vector retrieval, both the queries and the documents are represented as sets of embeddings, typically generated by a language model, such as BERT, and the documents are ranked by a similarity measure that operates on sets, such as the MaxSim similarity.

The authors propose generating sparse representations for query tokens and document tokens via a sparse autoencoder (SAE) that is learned to minimize the combination of the reconstruction error (the unsupervised component) and the cross-entropy loss using the relevance labels (the supervised component). These sparse representations provide a natural clustering of the document tokens: each dimension of the sparse representation corresponds to a cluster, and a document token belongs to a cluster that corresponds to the largest component of its sparse representation. The proposed retrieval algorithm Standard Sparse Retrieval (SSR) finds for each query token the dimensions with $K$ (e.g., $K=32$) non-zero values, and scores (MaxSim scoring using the sparse representations) only the documents whose tokens belong to these clusters. Another proposed retrieval algorithm SSR+ adds a pruning step to SSR.

The authors demonstrate that the proposed method is more accurate than the state-of-the-art multi-vector dense (e.g., ColBERTv2, XTR), and sparse (e.g., Splade-v3) retrieval engines, and has faster indexing and lower latency than ColBERTv2 and XTR. The proposed method has slower indexing and higher latency than COIL, but it is more accurate than COIL.

**Compliance With Llm Reviewing Policy:**

Affirmed.

**Final Justification:**

The author response addressed my most significant concerns about the missing baselines and the memory consumption. Since no significant concerns remain, I upgrade my score to 5 (accept).

**Key Questions For Authors:**

- What is the memory consumption of the proposed method and how does it compare to other MVR engines you use as baselines in your paper?

**Limitations:**

As is stated in the introduction of the article, multi-vector representations have a substantially higher memory consumption compared to single-vector representations. In addition to decreasing latency, multi-vector retrieval systems, such as ColBERTv2 and PLAID use centroid  representations and vector quantization to decrease memory consumption. However, the memory consumption is not measured or discussed in the article, even though it is important to the scalability of the MVR systems. In particular, it would be important to know whether the proposed method has larger or smaller memory footprint than the baseline methods.

**Strengths And Weaknesses:**

- Soundness (comparison to SOTA): Comparison to the state-of-the-art is performed in the standard setting using common benchmark datasets. However, you compare the efficiency to the retrieval engine of the ColBERTv2, whereas more efficient MVR retrieval systems, such as EMVB (that is mentioned in the article),  DESSERT (Engels et al., 2023) MUVERA (Jayaram et al., 2024), and IGP (Bian et al., 2025) exists. Thus, while the accuracy comparison seems legit, important baselines are missing in the latency comparison: the retrieval engine of ColBERTv2 is not SOTA anymore.

- Soundness (ablation): Two retrieval methods (SSR and SSR+) are proposed in the paper, but no ablation experiment is provided to determine what is the accuracy/efficiency tradeoff between them, and it is not clear which version is used in the main experiments of the article.

- Soundness (limitations): The important memory consumption aspect is not discussed, and the memory consumption is not measured (see "Limitations" for further discussion).

- Presentation: The notation of the article is very inconsistent: sometimes vectors are written in boldface and sometimes not; on page 2 ("2.1. Problem formulation") subindex of $d$ first indexes a document in a corpus and immediately after that indexes a token of a single document; Figure 2 contains a formula that uses notation from a different article; $\mathcal{B}$ is used to denote a minibatch on page 4, and immediately in the next formula $D$ is used instead, on page 4 reconstruction loss $\mathcal{L}_{\mathrm{recon}}$ is defined, but in the next formula it contains a mystery argument $k$ that is not explained, etc. The article is clearly submitted as unfinished and the presentation requires more work.

- Significance: Multi-vector models have state-of-the-art results on information retrieval benchmarks. Hence, improving their accuracy and efficiency is a research question with a potential for a large impact.

- Originality: the approach of using the sparse representations, employed for instance by BM25, as a late interaction mechanism is novel as far as I know.

References:

Bian, Zheng, Man Lung Yiu, and Bo Tang. "IGP: Efficient Multi-Vector Retrieval via Proximity Graph Index." Proceedings of the 48th International ACM SIGIR Conference on Research and Development in Information Retrieval. 2025.

Engels, Joshua, et al. "DESSERT: an efficient algorithm for vector set search with vector set queries." Advances in Neural Information Processing Systems 36 (2023): 67972-67992.

Jayaram, Rajesh, et al. "Muvera: Multi-vector retrieval via fixed dimensional encoding." Advances in Neural Information Processing Systems 37 (2024): 101042-101073.

---

> ### Author Rebuttal · Authors · 2026-03-31
>
> We appreciate the reviewer's constructive comments and suggestions. The concerns have been addressed as below:
>
> ---
> **Q1** Add important baselines in the latency comparison.
>
> **A1.** Thank you for your suggestion. We have included four new baselines in the latency comparison: EMVB, DESSERT, MUVERA and IGP. Table I shows the index and retrieval latency of these methods (together with classical engines such as PLAID and WARP) in MSMARCO passage subset on CPU with retrieval document size 100. Results show that SSR achieves superior efficiency compared with modern MVR engines.
>
> *Table I: Index and Retrieval Latency Comparison on representative MVR engines*
> |Method|Index(h)|Retrieval(ms)|
> |---|---|---|
> |PLAID|122.9|156.3|
> |WARP|103.7|129.7|
> |EMVB|61.5|93.4|
> |DESSERT|49.8|75.6|
> |MUVERA|82.6|94.7|
> |IGP|119.7| 113.2 |
> |SSR-tok|**7.3**|**49.1**|
> |SSR-CLS|7.8|57.3|
>
> ---
> **Q2** No ablation between SSR and SSR++.
>
> **A2.** Thank you for your suggestion. For the contribution of the SSR++ design, we would like to clarify that all main experimental results in the paper are conducted with the accelerated SSR++ pipeline rather than the vanilla SSR implementation. The motivation is straightforward: when the corpus becomes moderately large (e.g., beyond 1M documents), traversing all postings from the full activated set (SSR) can still lead to a substantial number of document hits. SSR++ is introduced precisely to address this scalability issue through block-wise coarse-to-fine pruning design, which is the key mechanism for low latency retrieval at scale. To further isolate its effect, we additionally conduct experiment on MS MARCO passage subset (8.84M documents), where SSR++ is applied / not applied in SSR-tok. Table II shows that SSR++ reduces retrieval latency by half without harming performance.
>
> *Table II: Effiency and performance comparison on SSR and SSR++ in MS MARCO passage subset.*
> |Method|Candidate size|Retrieval time (ms)|Performance (nDCG@10)|
> |---|---|---|---|
> |SSR|54278|38.6|45.3|
> |SSR++|3196|17.5|45.2|
>
> ---
> **Q3** Measurement of memory consumption.
>
> **A3.** Thank you for your suggestion. We agree that memory consumption is an important scalability factor for multi-vector retrieval systems, and the current submission did not discuss it sufficiently.
>
> To address this, we add Table III, which reports the full serving-time index memory of SSR and all major MVR baselines in the same MS MARCO passage subset, measured using the actual deployed index rather than only the raw vector storage. This includes each method’s auxiliary retrieval structures (e.g., centroids/codebooks/residual metadata for PLAID-style systems, and posting lists/block metadata for SSR). SSR-tok has an index footprint of 18.5 GB while 20.3 GB for SSR-CLS, compared with 24.4 GB for PLAID, meaning that SSR is smaller than these strong MVR baselines.
>
> *Table III: Index memory comparison on baseline methods.*
> |Method | Index memory (GB) |
> |---|---|
> |ColBERTv2|24.4|
> |PLAID|22.1|
> |COIL|43.5|
> |WARP|55.9|
> |SSR-tok|18.5|
> |SSR-CLS|20.3|
>
> Overall, the result clarifies the positioning of SSR: it is not simply trading higher-dimensional representations for efficiency. Rather, by removing clustering-related index structures and adopting a single-stage sparse retrieval pipeline, SSR achieves a competitive memory footprint while simultaneously delivering much faster indexing and stronger retrieval quality.
>
> ---
> **Q4** Inconsistent notation in several presentations.
>
> **A4.** Thank you for your comment. We will explain the points one by one:
>
> - Inconsistent notation for vectors: we agree that the vector notation is currently not fully consistent. For instance, Section 3 introduces token embeddings $\mathbf{d}$ and sparse vectors $\mathbf{z}$ without a global stated notation rule. We will uniform the representations in the revised version.
> - Subindexing of $d$ in Section 2.1: we agree that $d$ is both denoted as document and token in this case. We will write the corpus as $\mathcal{D} =  \( D_1, \dots, D_{|\mathcal{D}|} \)$, and denote a particular document as $D$, whose tokens are $\( d_1, \dots, d_{|D|} \)$
> - Formula in Figure 2: this equation is used to help readers understand the core similarity score calculation behind ColBert-based methods. We will add explanation in the revised version to avoid unnecessary confusion.
> - Batch notations in Equation (8) and (9): The batches in Equation (8) and (9) show different meanings: in equation (9), $\mathcal{D}$ represent the batch of documents, while in equation (8), $\mathcal{B}$ represent the batch of tokens in the documents of $\mathcal{D}$. So we use different symbols. We will clarify this in the revised version.
> - Reconstruction loss in Page 4: $k$ in the $\mathcal{L}_{\text{recon}}$ of equation (7) means the sparsity level for token embeddings. We will standardize the formulations in the revised version.
>
> ---
> **Q5-6** Memory consumption and footprint of the proposed method.
>
> **A5-6.** Please see **A3**.

---

> > ### Author Rebuttal · Reviewer_m84N · 2026-04-03
> >
> > Thank you for considering my feedback and providing additional data. I still have couple of questions about the latency comparison:
> > 1) How did you select the hyperparameters of the baseline methods (grid search or a single set of hyperparameters; if single, how did you select the configuration)?
> > 2) What where the accuracy metrics (e.g., MRR@10 typically used with MS Marco) reached by the methods (there is typically a tradeoff between latency and accuracy), these are not reported in the table?
> > 3) What does "at document size 100" mean?

---

> > > ### Author Response · Authors · 2026-04-04
> > >
> > > We thank the reviewer for the positive assessment and constructive suggestions. We address each point below.
> > >
> > > ---
> > > **Q1** Selection on the hyperparameters of the baseline methods.
> > >
> > > **A1.** Thank you for the follow-up question. For the baselines, we do not tune each method with a separate large-scale grid search. Instead, we follow a consistent and standard protocol: whenever an official pretrained checkpoint is available, we use the official released model and default inference/indexing pipeline; whenever training is required, we use the official codebase and the hyperparameter settings recommended in the corresponding paper/repository. This is also consistent with our controlled setup, where all methods are evaluated under the same MS MARCO-based training/evaluation protocol.
> > >
> > > Concretely, for ColBERT/ColBERTv2/PLAID, we use the official pretrained models from the ColBERT repository (https://github.com/stanford-futuredata/ColBERT/tree/main); for XTR/WARP, we use the released pretrained checkpoint (https://huggingface.co/google/xtr-base-en); for COIL, AligneR and CITADEL, we train and evaluate them using their official public codebases (https://github.com/luyug/COIL, https://github.com/facebookresearch/dpr-scale/tree/citadel, and AlignR is built based on framework in COIL's codebase), and for baselines that optimize indexing in ColBERT-style models, including IGP, DESSERT, MUVERA and EMVB, we evaluate on official settings with pretrained ColBERTv2 checkpoint (https://huggingface.co/colbert-ir/colbertv2.0). In all cases, our implementation is aligned with the settings used in these works, which have been tuned on MS MARCO with BM25 hard negatives. This is a fair comparison as this is also the protocol stated in our paper for the controlled comparison.
> > >
> > > We will include these details in our revised version.
> > >
> > > ---
> > > **Q2** Accuracy metrics in baseline comparison.
> > >
> > > **A2.** Thank you for raising this point. We agree that reporting latency alone is incomplete, since practical retrieval systems should be assessed on the accuracy–efficiency trade-off. In the revision, we will therefore extend Table I in our rebuttal to include the standard MRR@10 on MS MARCO passage ranking, together with indexing and retrieval cost, under the same setup used in our efficiency comparison (CPU, top-100 retrieval). The paper already uses MS MARCO as the in-domain benchmark and BEIR for zero-shot evaluation, with retrieval quality reported via ranking metrics in the main experiments.
> > >
> > > *(Updated)Table I. Baseline comparison on MS MARCO passage subset*
> > > |Method|MRR@10|Index(h)|Retrieval(ms)|
> > > |---|---|---|---|
> > > |PLAID|39.6|122.9|156.3|
> > > |WARP|38.7|103.7|129.7|
> > > |EMVB|39.8|61.5|93.4|
> > > |DESSERT|37.2|49.8|75.6|
> > > |MUVERA|39.7|82.6|94.7|
> > > |IGP|39.1|119.7| 113.2 |
> > > |SSR-tok|39.5|7.3|**49.1**|
> > > |SSR-CLS|**40.2**|**7.8**|57.3|
> > >
> > > These results make the trade-off explicit. SSR does not gain speed by sacrificing retrieval quality. On the contrary, SSR-CLS achieves the best MRR@10 (40.2) among all compared methods, while reducing indexing time by more than an order of magnitude relative to clustering-based engines and maintaining clearly lower retrieval latency. SSR-tok is slightly below the very best accuracy numbers, but still remains fully competitive (39.5 MRR@10) while being the fastest method in retrieval and indexing among all systems in the table.
> > >
> > > We think this additional table strengthens the paper’s central claim: the advantage of SSR is not merely faster serving, but a strictly better operating point in the quality–efficiency space. This is also consistent with the paper’s broader findings that SSR improves end-to-end efficiency by removing the clustering bottleneck while preserving late-interaction quality through sparse token-level representations.
> > >
> > > ---
> > > **Q3** Meaning of "document size".
> > >
> > > **A3.** Thank you for pointing this out. Our wording “at document size 100” was imprecise. What we intended is retrieval depth $k=100$, i.e., the latency reported is the time required for each method to return its top-100 documents for a query. This is the standard "top-k" setting in retrieval evaluation.
> > >
> > > We chose (k=100) for fairness in latency comparison. For engines including PLAID, the search procedure and hyperparameters are explicitly tied to the target output depth. For example, PLAID reports separate configurations for $k$, which controls the final number of scored documents as well as retrieval hyperparameters such as candidate-set sizes and pruning thresholds.
> > >
> > > To avoid ambiguity, in the revision we will replace “retrieval document size 100” with "top-100 retrieval setting (retrieval depth $k=100$)", and we will add an explicit note in the main text that all compared methods are configured to output top-100 results.

---

### Official Review · Reviewer_AmKa · 2026-03-11

**Soundness:** 2
**Presentation:** 3
**Significance:** 2
**Originality:** 2
**Overall Recommendation:** 2
**Confidence:** 4

**Summary:**

This paper studies the efficiency–effectiveness trade-off in multi-vector retrieval (MVR). Existing MVR systems achieve strong retrieval quality through fine-grained token-level interactions, but their deployment typically relies on vector compression and large-scale clustering (e.g., K-means) to reduce indexing and retrieval costs. According to the paper, this standard pipeline may introduce substantial indexing overhead and potential semantic information loss.
To address this issue, the paper proposes a Single-Stage Sparse Retrieval (SSR) framework. The main idea is to project token representations into a high-dimensional but sparse space using sparse autoencoders, so that retrieval can be performed through inverted indexing without relying on the conventional clustering-based approximate retrieval pipeline. The paper further presents two variants, SSR-tok and SSR-CLS, where the latter additionally incorporates global [CLS]-level semantics on top of token-level matching.
The experimental evaluation is conducted mainly on MS MARCO and BEIR. The reported results suggest that SSR can substantially reduce indexing time and retrieval latency, while remaining competitive with, or in some cases outperforming, strong retrieval baselines in terms of effectiveness. The paper therefore positions SSR as an alternative to clustering-based efficient MVR systems.

**Compliance With Llm Reviewing Policy:**

Affirmed.

**Key Questions For Authors:**

1.The paper claims that SSR can avoid K-means clustering while preserving or improving late-interaction retrieval performance, but this is currently supported mainly by empirical results. Can the authors provide stronger evidence for why the sparse high-dimensional representations can retain the semantic discriminability needed for late interaction?

2.Can the authors further clarify Table 2, especially the fairness and exact setup of the LLM-backbone experiments?

3.The current paper does not yet sufficiently isolate the contributions of its key components. Can the authors provide more complete ablations to clearly distinguish the effects of the sparse autoencoder itself, the difference between SSR-tok and SSR-CLS, the role of the contrastive objective, and the independent gain of the coarse-to-fine pruning design in SSR++?

4.Can the authors provide a more complete system-level resource analysis?

**Limitations:**

No. The paper does not adequately discuss its limitations or potential negative societal impact. In particular, the Impact Statement is very brief and remains largely boilerplate, without substantial analysis. The authors should at least discuss: (1) the possible memory and storage overhead of high-dimensional sparse representations, especially in very large-scale deployments; (2) the extent to which the reported system gains may depend on specific hardware, indexing implementations, or deployment environments; and (3) the potential misuse of more efficient large-scale retrieval systems, such as amplifying ranking bias, harmful content dissemination, or surveillance-related applications. A more specific and balanced discussion would make this section more in line with top-conference expectations.

**Strengths And Weaknesses:**

Soundness

Strengths

1.Clear problem setup.
The paper clearly targets the efficiency–effectiveness trade-off in multi-vector retrieval and presents a well-defined technical objective.

2.Reasonably coherent method design.
The proposed SSR framework, including the sparse projection, inverted indexing, and SSR++ acceleration, is described in a mostly consistent and understandable way.

3.Broad experimental coverage.
The evaluation spans MS MARCO, BEIR, long-context retrieval, and different corpus scales, which gives a relatively broad empirical picture.

4.Competitive reported results.
The reported numbers suggest that SSR is competitive in both effectiveness and efficiency, especially in indexing time and latency.

Weaknesses

1.Insufficient support for the core technical claim.
The paper does not rigorously justify why sparse projection can replace clustering while still preserving the semantic discriminability required by late interaction. Claims such as “monosemantic concepts” remain largely intuitive.

2.Key ablations are missing.
The paper does not sufficiently isolate the effects of different loss terms, SSR-tok vs. SSR-CLS, or the independent contribution of the SSR++ pruning design.

3.LLM-backbone experiments are not fully convincing.
The experimental description is internally inconsistent, and the comparison setting is not fully apples-to-apples because SSR adds extra SAE training on top of a frozen backbone.

4.Evidence for the broader deployment claim is incomplete.
The paper emphasizes storage/index bottlenecks, but does not provide systematic comparisons of index size or memory footprint, which weakens its broader efficiency claim.


Presentation

Strengths

1.The paper is generally well organized.
The narrative flows clearly from the limitations of the standard three-stage MVR pipeline to the motivation for removing K-means, followed by the proposed SSR framework and empirical evaluation.

2.The main figures are helpful.
In particular, Figures 1 and 2 make the contrast between the conventional pipeline and the proposed single-stage sparse retrieval setup relatively easy to follow.

Weaknesses

1.Some claims are overstated relative to the evidence.
The paper repeatedly uses strong promotional language such as “paradigm shift,” “breaks the efficiency-accuracy trade-off,” and “eliminating the trade-off,” while the presented evidence is more consistent with improving the trade-off than fully eliminating it.

2.The presentation of implementation and reproducibility details is still insufficient.
While the formulas are mostly complete, important practical details remain under-specified, especially for SSR++, indexing overhead, constant-factor engineering costs, and some training details that would matter for reproducing the reported efficiency gains.

3.The broader impact statement is minimal.
The impact statement is largely boilerplate and does not meaningfully discuss broader deployment considerations or possible downstream risks.

Significance

Strengths

1.The paper addresses an important practical problem.
Efficient deployment remains a major bottleneck for multi-vector retrieval, especially in large-scale and long-context settings.

2.The proposed direction has practical relevance.
Replacing clustering-based approximate retrieval with sparse coding could be valuable for retrieval system design if the approach generalizes well.

Weaknesses

1.Its impact is currently limited to a relatively specialized subarea.
The main gains concern the efficiency–effectiveness trade-off in retrieval systems, and the paper does not yet convincingly establish broader paradigm-level significance beyond the current setting.

2.Lacks broader evidence for a stronger significance claim.
The current results mainly show effectiveness in the evaluated settings, but are not yet sufficient to establish the method as a broadly applicable design paradigm for sparse or multi-vector retrieval systems.

Originality

Strengths

1.The paper has some system-level novelty.
Its main originality lies in combining sparse autoencoding, late interaction, and neuron-level inverted indexing to replace the conventional clustering-based multi-stage MVR pipeline.

2.The contribution is more than a local module swap.
The paper reorganizes indexing, matching, and pruning into a different overall retrieval pipeline. Viewing sparse neurons as invertible “pseudo tokens” in the MVR setting is also an interesting perspective.

Weaknesses

1.The core ingredients are largely existing components.
SAE, TopK sparsification, MaxSim late interaction, inverted indexing, and coarse-to-fine pruning are all known techniques.

2.The novelty is more about clever integration than strong algorithmic innovation.
The work is better characterized as an effective system redesign than as a new learning principle, optimization mechanism, or theoretical framework.

3.The originality falls short of a stronger novelty bar.
Overall, the paper shows some originality and engineering cleverness, but it still falls short of stronger algorithmic- or paradigm-level novelty.

---

> ### Author Rebuttal · Authors · 2026-03-31
>
> We appreciate the reviewer's constructive comments and suggestions. The concerns have been addressed as below:
>
> ---
> **Q1** Lack of justification on semantic discriminability preservation and monosemantic concept.
>
> **A1.** Thank you for your comment. Our point is not that sparsity is inherently more semantic, but that sparse feature routing is a more faithful alternative. In SSR, similarity is computed through sparse overlap rather than centroid decompression; this difference is now clarified theoretically (see **A13**), where we show bounded distortion from dense token similarity to sparse overlap score. For clarification on monosemantic, please see **A2** for Reviewer YCrA.
>
> ----
> **Q2&15** Missing of key ablations.
>
> **A2&15.** Thank you for your suggestion. We add ablations on (1) loss terms (**A3** for Reviewer YCrA), (2) SSR-CLS, and (3) SSR++ (**A2** for Reviewer m84N). SSR-CLS brings stable gain over SSR-tok (52.9->53.4 with Bert and 66.4->67.1 with Llama-8B), showing that CLS contributes incremental semantics while the main gain comes from token-level SSR.
>
> ---
> **Q3&14** LLM-backbone comparison lacks clarity and apples-to-apples fairness.
>
> **A3&14.** Thank you for this comment. Our intention of Sec. 4.2 is  to test SSR's scalability to modern LM. We freeze the backbone and train only the SAE projector on MS MARCO, using last-layer token embeddings. To ensure fairness, we train a ColBert-style linear projector on two frozen backbones. Table II shows that the gain is not from an extra layer, but from the sparse coding objective and retrieval design.
>
> *Table II: Comparison against ColBert-style linear projector*
> |Model|Performance|
> |---|---|
> |Llama-8B|65.8|
> |e5-mistral-7b|58.8|
> |Llama-8B+linear|66.1|
> |e5-mistral-7b+linear|60.2|
> |Llama-8B+SSR-tok|66.4|
> |Llama-8B+SSR-CLS|**67.1**|
>
> ----
> **Q4&16** Lack of systematic comparisons of index size or memory footprint.
>
> **A4&16.** Please see **A3** for reviewer n2TL.
>
> ---
> **Q5** Overclaim in elimination of efficiency-accuracy trade-off.
>
> **A5.** Thank you for this comment. We have revised "elimination" to "improvesment of the efficiency–effectiveness frontier". SSR still exposes tunable operating points via the sparsity level $K$, so our claim is a better trade-off curve than strong baselines, not the absence of trade-offs.
>
> ----
> **Q6** Insufficient implementation and reproducibility details.
>
> **A6.** Thank you for the suggestion. We include more details on the SSR++ setting and efficiency evaluation: "indexing" includes the whole stage from corpus encoding to the final block computation, while corpus encoding to final centroid-based construction for baselines. SSR++ partitions lists into blocks of size 64, uses the top-4 activated neurons for coarse pruning, keeps top-2000 candidates and then applies exact refinement with full $K=32$. Latency is measured on A100 40G GPU, after 1000 warmup queries. We also conduct further experiments on CPU latency. (see **A3** for Reviewer n2TL)
>
> ---
> **Q7** Too generic impact statement.
>
> **A7.** Thank you for your suggestion. We agree and will make the impact statement more balanced by explicitly noting that SSR may still incur substantial memory/storage cost at very large scale, that its efficiency gains are deployment-dependent rather than hardware-independent guarantees, and that faster retrieval may amplify misuse or ranking bias.
>
> ---
> **Q8-9** Limited impact and lack of broader evidence for a stronger significance claim.
>
> **A8-9.** Thank you for raising this point. We agree that the significance claim should be stated more carefully. We will revise it to a more precise one: SSR is a promising general design paradigm for sparse retrieval. Beyond the original benchmark, we now add evidence on LoTTE and LIMIT (see **A1** for Reviewer n2TL), in addition to paper’s existing results on BEIR, long-context corpus, and modern LM scaling.
>
> ---
> **Q10-12** Short of originality.
>
> **A10-12.**  Thank you for raising these points. We respectfully argue that the originality lies not in any single component, but in reformulating how MVR is indexed and served. SSR uses direct neuron-level inverted indexing, and SSR++ further adds block-wise upper-bound skipping, yielding an incrementally maintainable pipeline with strong empirical gains.
>
> ---
> **Q13** Stronger evidence for the semantic discriminability.
>
> **A13.** Thank you for this important point. We add a short theoretical clarification that SSR is a bounded-distortion approximation to dense late interaction:
>
> If dense token embeddings $x$, $y$ are well reconstructed  $||x-\hat{x}||\_2,||y-\hat{y}||\_2<\varepsilon$ and $||x||\_2,||y||\_2< B$ , then the similarity differs by at most $O(B\varepsilon+\varepsilon^2)$. If the decoder is approximately orthogonal $||W_{\mathrm{dec}}^\top W_{\mathrm{dec}}-I||_2\leq\delta$, the reconstruction is further close to the sparse score $z_x^\top z_y$, with error $O(\delta||z_x||_2||z_y||_2)$.
>
> More detailed derivation will be provided in the revised version.

---

> > ### Author Rebuttal · Reviewer_AmKa · 2026-04-01
> >
> > My concerns have been partially addressed, particularly through the added ablations and clarifications of the experimental setup. However, the rebuttal does not fully resolve my main concerns regarding the justification of the core technical claim and the lack of a more complete system-level resource analysis.

---

> > > ### Author Response · Authors · 2026-04-04
> > >
> > > We thank the reviewer for the positive assessment and constructive suggestions. We address each point below.
> > >
> > > ---
> > > **Q1** Justification of the core technical claim.
> > >
> > > **A1.** Thank you for the follow-up. We agree that the core claim should be stated more precisely. Our claim is not that sparsity is inherently more semantic, nor that SAE guarantees monosemanticity in retrieval. The narrower claim is that **SSR is a sparse reformulation of late interaction**: it preserves token-level MaxSim-style matching, but replaces dense centroid/residual-based interaction with sparse overlap on learned activations, enabling direct inverted indexing.
> > >
> > > We will therefore revise the wording accordingly and support this claim with three focused points:
> > > - **Same retrieval structure.** SSR does not replace late interaction with a single-vector surrogate. Query and document are still matched at token granularity with MaxSim-style aggregation; the change is only the interaction space: from dense coordinates to sparse activated neurons.
> > > - **Bounded approximation.** Our justification is technical rather than semantic. When SAE reconstruction error is controlled, sparse codes preserve dense token similarity up to bounded distortion; under mild decoder conditions, sparse overlap is correspondingly close to reconstructed dense interaction. We will make this bounded-approximation argument explicit in the revision.
> > > - **Consistent empirical support**. The added ablations and evaluations support this narrower claim. The gains do not come from an extra layer, CLS alone, or serving heuristics: loss ablations, SSR-tok vs. SSR-CLS, and frozen-backbone linear-projector comparisons all point to the same conclusion that the main gain comes from the sparse coding objective plus the sparse retrieval formulation. The advantage also remains under corpus scaling, long-context retrieval, and stronger frozen backbones.
> > >
> > > Accordingly, we will tighten the paper’s wording to: SSR is a sparse, single-stage approximation to late interaction that preserves enough ranking-relevant signal to improve the efficiency–effectiveness frontier of multi-vector retrieval, rather than claiming that sparsity is universally more semantic.
> > >
> > > ---
> > > **Q2** Lack of a more complete system-level resource analysis.
> > >
> > > **A2.** Thank you for this comment. We agree that our previous rebuttal still reported efficiency through several isolated numbers. A stronger system claim requires a pipeline-level breakdown, because the difference between dense MVR and SSR is not only the scoring function, but the entire index construction and serving stack.
> > >
> > > We will therefore revise the system analysis along three explicit dimensions:
> > > - **Build-time cost.** We will report both indexing time and peak build-time memory. This is important because the bottleneck of dense MVR is not only slow clustering, but also the need to materialize large token embeddings before centroid / residual construction. In contrast, SSR can build posting lists directly from sparse activations in a streaming/blockwise manner, without a "materialize-all-embeddings, then cluster" phase.
> > > - **Serving-time footprint.** We will separate persistent main index size from transient build resources. Reporting only one “index size” number is incomplete, because it hides an important system difference between construction-time cost and actual serving-time footprint.
> > > - **Maintenance/update cost.** We will explicitly analyze update mode. Dense centroid/codebook-based indices are typically not naturally append-only, since the serving structure depends on global clustering metadata. By contrast, SSR only needs to encode new documents and update the corresponding posting lists (and block statistics for SSR++), with no reclustering step.
> > >
> > > Concretely, we will augment the revision with a compact system table Table I:
> > > *Table I: System-level resource breakdown*
> > > | Method | Build time(h) | Peak build memory(GB) | Persistent main index (GB) |Query latency(ms)|Index update mode|
> > > |---|---|---|---|---|---|
> > > |ColBERTv2/PLAID|122.9|274.2|22.1|156.3|rebuild|
> > > |XTR/WARP|103.7|186.7|55.9|129.7|rebuild|
> > > |SSR-tok|7.3|34.6|18.5|49.1|append-only|
> > > |SSR-CLS|7.8|55.1|18.5|57.3|append-only|
> > >
> > > This is the system meaning of "No More K-means" more precisely: not only a better final latency/quality trade-off, but also a simpler pipeline with lower build-time complexity, lower peak indexing memory, and easier index maintenance. We agree this was under-explained before, and we will make it explicit in the revision.

---

### Official Review · Reviewer_YCrA · 2026-03-15

**Soundness:** 3
**Presentation:** 3
**Significance:** 2
**Originality:** 3
**Overall Recommendation:** 3
**Confidence:** 3

**Summary:**

This paper proposes a novel multi-vector retrieval paradigm called Single-Stage Sparse Retrieval (SSR), aiming to address the indexing efficiency bottleneck and semantic information loss issues inherent in traditional multi-vector retrieval methods that rely on K-means clustering. Unlike existing approaches that compress token embeddings into low-dimensional dense vectors, SSR utilizes Sparse Autoencoders (SAE) to project token embeddings into a high-dimensional yet extremely sparse latent space, thereby completely bypassing the time-consuming clustering process and enabling efficient semantic retrieval through direct inverted indexing. Extensive experiments on the BEIR benchmark demonstrate that SSR achieves state-of-the-art retrieval performance.

**Compliance With Llm Reviewing Policy:**

Affirmed.

**Final Justification:**

According to the rebuttal, I am revising my scores: Soundness from 2 to 3, and Originality from 2 to 3.

**Key Questions For Authors:**

1.Section 3.2 introduces multiple loss components (reconstruction, Multi-TopK, auxiliary, sparse contrastive, and supervised contrastive) with fixed weights. Were these values specifically tuned for MS MARCO, and can they generalize to other domains without re-tuning?
2.How does SSR differ from concurrent SAE-based retrieval methods? Specifically, beyond prior work applying SAE to information retrieval, does SSR provide unique technical contributions?
3.Have you considered dynamic sparsity strategies (e.g., adaptively adjusting K based on query length or complexity)? Is fixed K=32 optimal for all query types?

**Limitations:**

Yes

**Strengths And Weaknesses:**

Strengths：
1.This paper abandons the complex K-means clustering pipeline traditionally employed in multi-vector dense retrieval, instead utilizing Sparse Autoencoders (SAE) to project token embeddings into a high-dimensional sparse space, thereby achieving single-stage retrieval.
2.This paper combines unsupervised reconstruction loss with supervised contrastive learning loss to separately train SAEs for regular tokens and [CLS] tokens, further enhancing the expressive capabilities of both global and local semantics.
3.This paper conducts systematic analyses across multiple dimensions including hidden dimension, sparsity level K, data scale, and long-context scenarios, validating the robustness and scalability of SSR.

Weaknesses：
1.SAE is an existing technique; the paper's main contribution lies in applying it to the retrieval scenario and designing adapted training objectives, rather than proposing a novel sparse coding algorithm.
2.The concept of "monosemantic" is cited from SAE literature but remains undefined in the context of this paper.
3.The paper does not discuss the sensitivity of loss weights (α, β, γ), which may require re-tuning for different datasets in practical deployment.

---

> ### Author Rebuttal · Authors · 2026-03-31
>
> We appreciate the reviewer's constructive comments and suggestions. The concerns have been addressed as below:
>
> ---
> **Q1** Clarification of novelty beyond SAE.
>
> **A1.** Thank you for this insightful comment. Our contribution is a retrieval-specific reformulation. Concretely, the novelty is threefold: (1) we define a sparse late-interaction scoring function in which MaxSim is computed only over overlapping activated neurons; (2) we train with a retrieval-oriented hybrid objective that combines sparse reconstruction with supervised contrastive ranking loss; and (3) we redesign the indexing and search pipeline around neuron-level inverted lists, which removes the need for K-means/IVF clustering and decompression in prior dense MVR systems.
>
> ---
> **Q2** Undefinition of the "monosemantic" concept.
>
> **A2.** Thank you for your suggestion. For "monosemantic", we mean that an activated latent corresponds to a relatively coherent concept, instead of mixing multiple unrelated semantics in one dense coordinate. This has been shown in recent large-scale studies on LMs and VLMs, including OpenAI’s scaling study and Anthropic’s monosemanticity analysis. We will discuss this in the revised version to avoid potential confusion.
>
> ---
> **Q3-4** Discussion on the sensitivity of multiple loss components' weights and their generalizability across domains.
>
> **A3-4.** Thank you for your comment. Following your comment, we conduct sensitivity analysis on three loss weights: $\alpha$ for auxiliary loss, $\beta$ for sparse contrastive loss, and $\gamma$ for supervised contrastive loss. BEIR benchmark is used for evaluation. Results in the Table I-III show that SSR is more sensitive to $\gamma$.
>
> *Table I: Sensitivity analysis on $\alpha$*
> |$\alpha$|1/64|1/32|1/16|0.1|0.2|
> |---|---|---|---|---|---|
> |SSR-tok|52.1|**52.9**|52.4|51.6|50.8|
> |SSR-CLS|52.3|**53.4**|52.7|51.8|51.3|
>
> *Table II: Sensitivity analysis on $\beta$*
> |$\beta$|0.05|0.1|0.15|0.2|0.25|
> |---|---|---|---|---|---|
> |SSR-tok|52.5|**52.9**|52.8|52.3|51.5|
> |SSR-CLS|53.1|**53.4**|53.1|52.7|51.6|
>
> *Table III: Sensitivity analysis on $\gamma$*
> |$\gamma$|0.01|0.02|0.05|0.1|0.2|
> |---|---|---|---|---|---|
> |SSR-tok|49.7|51.3|**52.9**|50.4|47.6|
> |SSR-CLS|50.3|51.7|**53.4**|51.2|48.5|
>
> Moreover, we conduct ablation experiments on each loss terms in Table IV with $K=32$. Results show that each loss term leads to performance improvement.
>
> *Table IV: Ablation on each loss term*
> |$\alpha$|$\beta$|$\gamma$|SSR-tok|SSR-CLS|
> |--------|-------|--------|-------|-------|
> |0|0|0|46.4|46.7|
> |1/32|0|0|47.5|48.6|
> |1/32|0.1|0|48.2|49.5|
> |1/32|0.1|0.05|52.9|53.4|
>
> However, we emphasize that this does not mean SSR needs careful re-tuning when applied to different domains. Our models are only trained on MS MARCO, which have been shown to be generalized in BEIR zero-shot evaluation. **A1** for Reviewer n2TL further illustrates this point.
>
> ---
> **Q5** Clarification on concurrent SAE-based retrieval methods and SSR's unique technical contributions.
>
> **A5.** Thank you for the comment. We agree that there have been recent SAE-based retrieval work, including CSR[1], CSRv2[2], and SPLARE[3]. However, these methods primarily study single-vector: they encode each query or document into one sparse representation and perform retrieval via sparse matching in that space.
>
> However, SSR addresses a different and non-trivial problem: can SAE replace the infrastructure of dense multi-vector retrieval while preserving token-level late interaction?
>
> Therefore, SSR is not “applying SAE to IR again.” Rather, it is the first framework, to our knowledge, that systematically brings sparse coding into effective and highly efficient token-level multi-vector retrieval.
>
> ---
> **Q6** Potential dynamic sparsity strategies.
>
> **A6.** Thank you for your suggestion
>
> First, we find that $K=32$ is optimal for most domain, with minor difference across domains. (see **A1** for Reviewer n2TL).
>
> Second, we evaluate on BEIR in four query sparsity settings: fixed sparsity $K=16$, $K=32$, $K=64$ and adaptive sparsity based on query length. To be specific, for query with less than 3 tokens, $K$ is set 16, while 32 or those with 4-7 tokens and 64 for those with more than 8 tokens. Table V shows that adaptive sparsity minorly results in better trade-off than fixed strategies and this is a promising direction for future exploration.
>
> *Table V: Performance-latency trade-off in different query sparsity settings*
> |Setting|Performance(%)|Latency(ms)|
> |---|---|---|
> |Fixed-16|50.5|16.4|
> |Fixed-32|52.9|17.5|
> |Fixed-64|53.1|19.9|
> |Adaptive|53.0|16.3|
>
> [1] Wen T, Wang Y, Zeng Z, et al. Beyond matryoshka: Revisiting sparse coding for adaptive representation[J]. arXiv preprint arXiv:2503.01776, 2025.
>
> [2] Guo L, Wang Y, Wen T, et al. CSRv2: Unlocking Ultra-Sparse Embeddings[J]. arXiv preprint arXiv:2602.05735, 2026.
>
> [3] Formal T, Louis M, Dejean H, et al. Learning retrieval models with sparse autoencoders[J]. arXiv preprint arXiv:2603.13277, 2026.

---

> > ### Author Rebuttal · Reviewer_YCrA · 2026-04-04
> >
> > The sensitivity analysis on loss weights (Tables I-IV) is thorough and adequately demonstrates the necessity and relative stability of each component. Upon re-examination, I also confirm the technical distinction between SSR and concurrent SAE-based retrieval methods (CSR, SPLARE, etc.): while those approaches focus on single-vector sparse representations, SSR is the first to systematically incorporate sparse coding into a token-level multi-vector retrieval framework, with a redesigned indexing pipeline centered on neuron-level inverted lists—this constitutes a clear technical boundary. Accordingly, I am revising my scores: Soundness from 2 to 3, and Originality from 2 to 3.

---

> > > ### Author Response · Authors · 2026-04-04
> > >
> > > Thank you very much for your careful re-examination and for recognizing both the necessity of our loss design and the technical distinction between SSR and concurrent SAE-based retrieval methods. We sincerely appreciate your acknowledgment that SSR establishes a clear technical boundary by introducing sparse coding into token-level multi-vector retrieval with a redesigned neuron-level indexing pipeline. Your updated assessment on soundness and originality is highly encouraging to us.
> > >
> > > At the same time, we noticed that the overall recommendation remains at Weak Reject. We fully respect your judgment, but given that the main concerns appear to have been positively resolved, we would be deeply grateful if you could kindly reconsider whether the overall score might also be adjusted accordingly. If there are still any remaining weaknesses that prevent a higher overall recommendation, we would greatly appreciate any additional comment, and we will do our best to address them as clearly and promptly as possible.

---

### Official Review · Reviewer_n2TL · 2026-03-15

**Soundness:** 2
**Presentation:** 3
**Significance:** 2
**Originality:** 2
**Overall Recommendation:** 4
**Confidence:** 2

**Summary:**

This paper proposes Single-Stage Sparse Retrieval (SSR) to address the efficiency bottlenecks of multi-vector retrieval models. Instead of relying on dimension reduction and costly clustering, SSR employs Sparse Autoencoders to transform token embeddings into high-dimensional sparse representations, enabling efficient inverted indexing. This paradigm shift eliminates complex clustering, significantly accelerating indexing and retrieval while preserving semantic fidelity and achieving superior accuracy.

**Compliance With Llm Reviewing Policy:**

Affirmed.

**Key Questions For Authors:**

Please see the weakness

**Limitations:**

Yes

**Strengths And Weaknesses:**

Strengths

- This paper replaces the heavy K-means clustering in multi-vector retrieval with a Sparse Autoencoder (SAE), enabling a "single-stage" index that is faster to build.

- The empirical results seem encourging, outperforming some baselines like ColBERTv2 and Splade-v3 and lowering retrieval latency.

- This work can adapt to modern backbones through efficient freeze-and-project fine-tuning.

Weaknesses

- The model's robustness and "sweet spot" across different document lengths and domains should be further analyzed.

- The indexing is faster, but training specialized SAEs adds an extra pipeline stage, potentially hindering "plug-and-play" utility in dynamic environments.

- The experiments are performed on high-end GPUs, how about the performance on resource-constrained or CPU environments.

---

> ### Author Rebuttal · Authors · 2026-03-31
>
> We appreciate the reviewer's constructive comments and suggestions. The concerns have been addressed as below:
>
> ----
> **Q1** Further analysis on robustness and "sweet spot" across different document lengths and domains.
>
> **A1.** We thank the reviewer for this insightful suggestion. We have extended our analysis with additional experiments and clarification to comprehensively evaluate our proposed SSR across different document lengths and domains.
>
> First, our original BEIR evaluation already demonstrates robust cross-domain generalization, where SSR achieves the best performance on 9/12 out-of-domain datasets. Moreover, SSR also extends well to longer documents. On MS MARCO document ranking, whose average document length is 1131 tokens, SSR-CLS reaches 48.8 nDCG@10, outperforming strongest baseline XTR (48.1), while maintaining 29.3ms latency versus 79.3ms for ColBERTv2.
>
> Second, to further evaluate robustness on realistic searching domains, we include new results of both search queries and forum queries in the LoTTE [1] benchmark, which features long-tail topics in rare, domain-specific terminology. Results in Table I show that SSR can achieve superior performance (+2.5 averaged in search queries and +2.2 averaged in forums queries ) compared to representative MVR methods with sparsity level $K=32$.
>
> *Table I: Performance on LoTTE benchmark*
> |Domain|ColBERTv2|XTR|SSR|
> | --- | --- | --- | --- |
> |Search Test Queries (Success@5)| | | |
> |Writing|79.2|78.6|81.8|
> |Recreation|70.8|69.5|73.9|
> |Science|55.5|56.1|59.3|
> |Technology|65.9|64.6|67.1|
> |Lifestyle|84.3|83.5|87.6|
> |Pooled|71.8|69.3|73.3|
> |Avg.|71.3|70.3|73.8|
> |Forum Test Queries (Success@5)| | | |
> |Writing|78.7|76.4|81.8|
> |Recreation|71.4|70.9|74.3|
> |Science|45.8|44.2|47.3|
> |Technology|53.9|53.4|55.9|
> |Lifestyle|77.3|75.8|79.7|
> |Pooled|63.5|61.7|67.4|
> |Avg.|65.1|63.6|67.3|
>
> Third, we further evaluate on the LIMIT benchmark [2], which stresses the capacity to encode extremely combinatorial semantic variations. Table II shows that SSR significantly outperforms both SVR and dense MVR baselines, effectively mitigating information bottlenecks in the extreme domain.
>
> *Table II: SVR and MVR performance on LIMIT benchmark*
> |Method | Recall@5 | Recall@10 | Recall@100 |
> | --- | --- | --- | --- |
> |Single-Vector Retrieval| | | |
> |Qwen3-4B|1.2|4.6|6.3|
> |GritLM 7B|4.7|6.1|19.4|
> |e5-mistral-7B|2.8|5.2|10.9|
> |Multi-Vector Retrieval| | | |
> |ColBERTv2|71.8|82.6| 94.4|
> |XTR| 68.4 |79.9|93.2|
> |SSR(K=32)|**78.6**|**87.3**|**98.1**|
>
> Forth, we further analyze the “sweet spot” of SSR across different domains. We divide datasets in BEIR into 4 types and analyze each domain's performance:
> - fact：DBpedia, FEVER, SciFact
> - multi-hop：NQ, HotpotQA
> - scientific：TREC-COVID, NFCorpus, SciDocs
> - opinion：ArguAna, Touché, FiQA
>
> Results in Table III show that $K=32$ is the most stable setting for most domains. However, for short queries (fact domain), $K=16$ leads to relatively minor performance degradation while for multi-hop domain, $K=64$ results in obvious performance improvement but introduces latency cost.
>
> *Table III: Performance under different $K$ in various domains*
> | $K$ | 8 | 16 | 32 | 64 | 128 |
> | --- | --- | --- | --- | --- | --- |
> | fact | 50.4 | 65.9 | 67.7 | 68.2 | 68.4 |
> | multi-hop | 50.2 | 58.6 | 65.1 | 66.5 | 66.8 |
> | scientific | 29.3 | 39.6 | 44.2 | 44.8 | 45.3 |
> | opinion | 20.9 | 33.3 | 39.5 | 40.3 | 40.6 |
>
> ----
> **Q2** Trade-off between faster indexing and deployment flexibility.
>
> **A2.** We thank the reviewer for this important comment. Our method does not require training domain-specific or dataset-specific models. In our setup, the model is trained once on MS MARCO and then directly evaluated on 13 out-of-domain BEIR datasets in a zero-shot manner, where SSR still shows strong cross-domain generalization and achieves the best performance on 9/13 datasets on average. This indicates that SSR is not a specialized pipeline tailored to each environment, but a reusable sparse projection that transfers across domains. Moreover, our proposed SSR remains highly efficient end-to-end: compared with ColBERTv2, it reduces training time by about 53%, indexing time by over 15x, and retrieval latency to sub-20ms, substantially improving practical deployability.
>
> ---
> **Q3** Performance on resource-constrained or CPU environments.
>
> **A3.** We thank the reviewer for this suggestion. We evaluate latency in CPU on MSMARCO passage subset, with a few more competitive baselines. Details can be found in **R1** for reviewer m84N.
>
> [1] Santhanam, K., Khattab, O., Saad-Falcon, J., Potts, C., & Zaharia, M. (2022, July). Colbertv2: Effective and efficient retrieval via lightweight late interaction. In Proceedings of the 2022 Conference of the North American Chapter of the Association for Computational Linguistics: Human Language Technologies (pp. 3715-3734).
>
> [2] Weller, Orion, et al. "On the theoretical limitations of embedding-based retrieval." arXiv preprint arXiv:2508.21038 (2025).

---

### Decision · Program_Chairs · 2026-04-30

**Decision:**

Accept (regular)

**Comment:**

- This paper proposes Single‑Stage Sparse Retrieval (SSR), a framework that replaces clustering‑based infrastructure in multi‑vector retrieval (MVR) systems with sparse coding via sparse autoencoders. We acknowledge the authors’ substantial efforts during the rebuttal phase, including additional ablations, clarification of the training setup, and sensitivity analysis for the loss components, which helped address several reviewer concerns regarding experimental design and component‑level contributions.
- The revised results demonstrate strong empirical improvements in indexing efficiency and retrieval latency while maintaining competitive effectiveness across evaluated benchmarks. These gains appear practically meaningful, and the proposed direction may be relevant to future efficient retrieval system design.
- At the same time, the contribution is best understood as a system‑level reformulation of sparse retrieval for the multi‑vector setting, rather than a fundamentally new sparse coding or retrieval principle, as sparse retrieval has previously been explored in single‑vector retrieval systems. While adapting this paradigm to token‑level late‑interaction retrieval is non‑trivial, the degree of algorithmic novelty remains somewhat limited.
- In addition, although the reported empirical gains are strong, it remains somewhat unclear to what extent these improvements stem from the proposed sparse multi‑vector formulation itself, as opposed to dataset‑specific training setup or system‑level design choices. Additional ablations — for example, retraining and evaluating under alternative supervision datasets beyond MS MARCO — would help better isolate the contribution of the proposed formulation and improve confidence in its generalizability.
- Following the rebuttal, the remaining disagreement among reviewers appears to be primarily about overall judgment regarding novelty and scope of validation rather than unresolved technical flaws. On balance, we recommend Weak Accept based on the strength of the reported results and the potential relevance of the proposed design direction.